# Interannual variability in Australia's terrestrial carbon cycle constrained by multiple observation types

Cathy M. Trudinger[1], Vanessa Haverd[2], Peter R. Briggs[2], and Josep G. Canadell[2]

[1]CSIRO Oceans and Atmosphere, Aspendale, Australia
[2]CSIRO Oceans and Atmosphere, Canberra, Australia

*Correspondence to:* Cathy Trudinger (cathy.trudinger@csiro.au)

**Abstract.** Recent studies have shown that semi-arid ecosystems in Australia may be responsible for a significant part of the interannual variability in the global concentration of atmospheric carbon dioxide. Here we use a multiple constraints approach to calibrate a land surface model of Australian terrestrial carbon and water cycles, with a focus on interannual variability. We use observations of carbon and water fluxes at 14 OzFlux sites, as well as data on carbon pools, litter-fall and streamflow. We include calibration of the function describing the response of heterotrophic respiration to soil moisture. We also explore the effect on modelled interannual variability of parameter equifinality, whereby multiple combinations of parameters can give an equally acceptable fit to the calibration data. We estimate interannual variability of Australian NEP of 0.12–0.21 PgC yr$^{-1}$ ($1\sigma$) over 1982-2013, with a high anomaly of 0.43–0.67 PgC yr$^{-1}$ in 2011 relative to this period associated with exceptional wet conditions following a prolonged drought. The ranges are due to the effect on calculated NEP anomaly of parameter equifinality, with similar contributions from equifinality in parameters associated with NPP and heterotrophic respiration. Our range of results due to parameter equifinality demonstrates how errors can be underestimated when a single parameter set is used.

## 1   Introduction

The growth rate of carbon dioxide ($CO_2$) in the atmosphere has significant interannual variability, mostly driven by variability in $CO_2$ uptake by terrestrial ecosystems (Rayner et al., 2008; Bastos et al., 2013; Le Quéré et al., 2015). Recent studies have shown that while mean terrestrial $CO_2$ uptake is dominated by tropical forests, the trend and interannual variability in terrestrial uptake are dominated by semi-arid ecosystems (Poulter et al., 2014; Ahlström et al., 2015; Liu et al., 2015). Uptake of $CO_2$ by land (net ecosystem production, NEP) is the balance between net primary production (NPP) and heterotrophic respiration, and NPP rather than heterotrophic respiration appears to be the main driving mechanism behind variability in the land sink (Poulter et al., 2014; Ahlström et al., 2015). Evidence from ecosystem models, atmospheric inversions and satellite observations (Poulter et al., 2014; Detmers et al., 2015) suggests that a strong land carbon sink in 2011 (Le Quéré et al., 2015; Bastos et al., 2013) was driven by growth of semi-arid vegetation in the southern hemisphere, with a large contribution from Australia associated with wet conditions of an extraordinary La Niña event following a prolonged drought.

Here we use the BIOS-2 implementation for Australia (Haverd et al., 2013a) of the Community Atmosphere Biosphere Land Exchange (CABLE) land surface model (Wang et al., 2010, 2011), described in detail in Sect. 2.1, to explore interannual variability (IAV) in Australian carbon fluxes between 1982 and 2013. This work builds on the study by Haverd et al. (2013b) that used BIOS-2 to estimate the mean carbon budget for Australia. We use a multiple constraints approach (Raupach et al., 2005) to optimise model parameters. We include some improvements to the model structure and forcing data (specifically, optimisation of the function describing the sensitivity of heterotrophic respiration to soil moisture, that has been shown by Exbrayat et al. (2013a) to be important for modelling NEP, and an improved product for vegetation cover). The implementation of the model described here is denoted BIOS-2.1.

We are also interested in the effect of uncertainty in model parameters on modelled IAV. It is now widely recognised in many areas of research that there is usually no single 'true' parameter set, but that there may be many parameter sets for a given model structure that are equally acceptable for simulating the data. There are different aspects to this. Firstly, two or more parameters may have a similar effect on model outputs, so can be difficult to distinguish. This is called equifinality (Aalderlink and Jovin, 1997; Beven, 2006; Tang and Zhuang, 2008) and leads to correlated errors in the estimates of model parameters. Whether parameters are uniquely identifiable from comparison of model outputs with observations will depend on how the model is formulated (over-parameterisation will increase the chances of equifinality), and what types of observations are used to calibrate the model. Different kinds of measurements, or measurements with information about processes acting on different timescales, will constrain different model parameters (Luo et al., 2009), so thought should be given to whether the available observations are likely to contain information about parameters of interest. Secondly, even without compensatory parameters, all models and the observations used to calibrate them are in error to some extent. Therefore, there is no reason to believe that the global optimum parameter set is more correct than other parameter sets that fit the observations to a lesser, but still acceptable, extent (Beven and Binley, 1992).

Although global search methods like Markov Chain Monte Carlo can usually do a more thorough search of parameter space than down-gradient methods, they are much more computationally expensive. Ziehn et al. (2012) showed that both can do a good job at obtaining the probability density function for parameters in a land surface model. Here we use a down-gradient search method to optimise a set of model parameters, with an efficient method (null-space Monte Carlo) to explore parameter equifinality.

The objectives of this study are to use multiple observation types to constrain the IAV of terrestrial carbon fluxes for Australia. Specifically, multiple observation types are used to optimise parameters in BIOS-2.1, by generating an ensemble of acceptable parameter sets that will allow us to see the effect of parameter equifinality. We then use these parameter sets in the model to calculate IAV in Australian NEP over recent decades. We are interested in the following questions: What is our best estimate of IAV in Australian carbon fluxes? How does parameter equifinality affect modelled estimates of IAV and the 2011 anomaly for Australia? How does parameter equifinality effect estimates of the processes contributing to IAV in NEP, including NPP and heterotrophic respiration and the effect of soil moisture on heterotrophic respiration? The outline of the paper is as follows. In Sect. 2 we describe the model, the forcing data, the observations used for calibration and validation, and the optimisation method. In Sect. 3 we present results, followed by discussion in Sect. 4 and conclusions in Sect. 5.

## 2 Methods

### 2.1 BIOS-2.1 model

Haverd et al. (2013a) described an implementation of the CABLE land surface model (Kowalczyk et al., 2006; Wang et al., 2011), CASA-CNP biogeochemical model (Wang et al., 2010) and Soil-Litter-Iso (SLI) soil model (Haverd and Cuntz, 2010) for Australia at fine spatial resolution ($0.05\,^{\circ} \times 0.05\,^{\circ}$ grid, which is roughly $5 \times 5$ km) using daily meteorology (downscaled for CABLE using a weather generator). This composite model and environment was referred to as BIOS-2, and makes use of the modelling environment built for the Australian Water Availability Project (AWAP) (King et al., 2009; Raupach et al., 2009). Modifications to CABLE, SLI and CASA-CNP for use in BIOS-2 are described in Haverd et al. (2013a). Some of the main features are that plant functional types are not used, instead each cell is partitioned into woody and grassy tiles. CABLE was run at an hourly timestep, and daily values of GPP, soil moisture and soil temperature were used to drive CASA-CNP at daily timesteps. Nitrogen and phosphorous cycles in CASA-CNP were disabled, and land management was not considered explicitly.

Here we extend this work as described in the following sections, with a new implementation for Australia denoted BIOS-2.1. We optimise model parameters for CABLE and CASA-CNP separately, using the method described in Sect. 2.4. The model parameters that are optimised are given in Tables 1 and 2. We run the model from 1900, but focus our analysis on the period between 1982 and 2013 for which we have consistent remotely sensed vegetation cover (Sect. 2.2).

### 2.1.1 Heterotrophic respiration function of soil moisture

Soil moisture has an important effect on heterotrophic respiration (Exbrayat et al., 2013a, b; Sierra et al., 2015). Soil moisture simulated by land surface models is known to be very model dependent (Koster et al., 2009), so a function of soil moisture that performs well in one model may not perform well in other models. The standard version of CASA-CNP uses a function from Kelly et al. (2000) for the dependence of heterotrophic respiration on soil moisture (Exbrayat et al., 2013b). The function is

$$f(\overline{s}) = \left( \frac{\overline{s} - 1.70}{0.55 - 1.70} \right)^{6.6481} \times \left( \frac{\overline{s} + 0.007}{0.55 + 0.007} \right)^{3.22} \tag{1}$$

where $\overline{s}$ is the root-mass weighted mean soil moisture content relative to saturated.

Here we optimise the heterotrophic respiration dependence on soil moisture, using a new function with six parameters. These parameters are optimised along with other model parameters in CASA-CNP. The form of the function was designed such that the function in Eq. 1 can be replicated with a particular choice of parameters, but the function also allows the type of behaviour

seen in many of the functions compared in Fig. 4c of Sierra et al. (2015). The equations for the function, with parameters q, c, $w_0$, $w_1$, $w_2$ and $w_3$ to be optimised, are as follows:

$$f(\bar{s}) = \begin{cases} \max \begin{cases} q \times \bar{s}^2 \\ \frac{0.5}{1-f(0)} \times \left[ 1 - \cos\left( \pi \frac{\bar{s} - w_0 + (c-1) \times w_1}{w_1 \times c} \right) - f(0) \right] \end{cases} & \text{if } \bar{s} < w_0 + w_1 \\ 1.0 & \text{if } w_0 + w_1 \leq \bar{s} \leq w_0 + w_1 + w_2 \\ 0.5 \left[ 1 + \cos\left( \pi \left( \bar{s} - w_0 - w_1 - w_2 \right) / w_3 \right) \right] & \text{if } \bar{s} > w_0 + w_1 + w_2 \end{cases} \quad (2)$$

where

$$5 \quad f(0) = 0.5 \times \left[ 1 - \cos\left( \pi \frac{0 - w_0 + (c-1) \times w_1}{w_1 \times c} \right) \right] \quad (3)$$

The function and parameters are shown in Fig. 1. The first part of the function (in red in Fig. 1a) is a quadratic with the rate of increase described by the parameter 'q'. The second part of the curve (green) is an increasing part of a cosine curve. The curvature is controlled by the parameter 'c' as it determines whether the full range of the cosine curve from -1 to +1, or just part of it near the top, is used. The solid and dashed green lines in Fig. 1a demonstrate the effect of parameter c, with

10 different values of c (1.0 and 5.0, respectively) but identical values of the other parameters. The width of this part of the curve is set by parameter '$w_1$'. As this function starts from zero at $\bar{s} = w_0$, in order to match up with the first (quadratic) part of the function, the maximum of the two functions is used where they overlap (the solid green line in Fig. 1a shows the maximum of the quadratic and cosine functions where they overlap, the dotted lines show the parts of the functions that are not used). If $w_0 + w_1$ hasn't already exceeded the maximum $\bar{s}$ value of 1.0, the next part of the curve (yellow) is a constant value of 1.0. The

15 width is set by parameter '$w_2$' and can be zero if no flat top is required. Then if $w_0 + w_1 + w_2 < 1.0$, the final part of the function (blue) is a decrease described by a cosine, with width '$w_3$'. If $w_0 + w_1$ is greater than 1.0, then $w_2$ is not used. Similarly, if $w_0 + w_1 + w_2$ is greater than 1.0, $w_3$ is not used. The function is continuous throughout the full range, but there is a discontinuity in the gradient when the quadratic meets the first cosine part of the function.

The parameters required for our function to approximately match the equation from Kelly et al. (2000) are (q,c,$w_0$,$w_1$,$w_2$,$w_3$)

= (0.0, 1.0, 0.025, 0.522, 0, 0.64). We use these as prior values in the parameter optimisation. The Kelly et al. (2000) function is shown by the dashed line in Fig. 1b. We chose not to optimise the temperature dependence of soil respiration because soil moisture (due to precipitation) has a much greater influence on interannual variability in soil respiration than temperature (see Supplementary Fig. S1). Exbrayat et al. (2013a) noted that the impact of the soil moisture response function on heterotrophic respiration is intimately connected with the skill of the land surface model to simulate soil moisture. It has been demonstrated

elsewhere (Frost et al., 2015; Holgate et al., 2016) that BIOS-2 performs well for soil moisture. Haverd et al. (2013a) manually adjusted the soil moisture dependence of soil respiration but did not include it in their formal parameter estimation.

## 2.2 Forcing data

The model is forced using gridded meteorological data, soil properties and vegetation cover at $0.05\,^{\circ} \times 0.05\,^{\circ}$ spatial resolution. The meteorological data and soil properties used here are as described in Haverd et al. (2013b). Briefly, they consist of daily meteorology from the Bureau of Meteorology's contribution to the Australian Water Availability Project (Grant et al., 2008; Jones et al., 2009), downscaled to hourly timesteps for CABLE using a weather generator, and soil properties taken from the McKenzie and Hook (1992) and McKenzie et al. (2000) interpretations of soil types mapped in the Digital Atlas of Australian Soils (Northcote et al., 1960, 1975).

In BIOS-2 (Haverd et al., 2013a, b), vegetation cover came from LAI derived from fPAR (fraction photosynthetic absorbed radiation) estimates obtained from the AVHRR and MODIS time series. These time series covered the periods 1990–2006 and 2000–2011, respectively. Here, vegetation cover is derived from the third generation (NDVI3g) of the GIMMS NDVI time series (Tucker et al., 2005; Zhu et al., 2013). This gives us a consistent vegetation cover timeseries over several decades (1982-2013). Total fPAR is partitioned into persistent (mainly woody) and recurrent (mainly grassy) vegetation components, following the methodology of Donohue et al. (2009) and Lu et al. (2003). This methodology takes advantage of low levels of seasonal change in LAI in woody vegetation, allowing seasonal variation in fPAR to be attributed principally to grassy vegetation. The remaining and relatively constant fPAR signal is attributed to woody vegetation. LAI for woody and grassy components are estimated by Beer's Law (e.g. Houldcroft et al., 2009):

$$LAI_W = -\frac{1}{k}\log_e\left(1 - fPAR_W\right) \tag{4}$$

$$LAI_G = -\frac{1}{k}\log_e\left(1 - \frac{fPAR_G}{1 - fPAR_W}\right) \tag{5}$$

where $W$ denotes the persistent or mainly woody vegetation type, $G$ denotes the recurrent or mainly grassy vegetation type and k is an extinction coefficient, set here to 0.5. In contrast to earlier BIOS-2 simulations (Haverd et al., 2013a, 2016b), Eq. 5 accounts for the effect of shading of grass by woody vegetation. We include one case without this correction for shading of grass, i.e. using

$$LAI_G = -\frac{1}{k}\log_e\left(1 - fPAR_G\right) \tag{6}$$

instead of Eq. 5, but all other results presented here use the shade correction.

## 2.3 Observations

The following observations were used for calibration of model parameters. We used monthly eddy flux data (evapotranspiration (ET), GPP and NEP) from 14 OzFlux sites (Beringer et al., 2016a; Isaac et al., 2016) listed in Table 3. The eddy flux data were processed using the DINGO (Dynamic INtegrated Gap filling and partitioning for OzFlux) methodology for processing raw flux tower data, as described in Donohue et al. (2014), Haverd et al. (2016a) and Beringer et al. (2016b). The period for which

we have observations at each site is shown in the 4th column of Table 3 (giving a total of 70 site-years of data). A subset of these monthly observations (40 site-years of data) was used for parameter estimation (column 5 of Table 3). We excluded eddy flux observations at Tumbarumba around 2003 because it is known that an insect attack combined with drought stress had a significant effect on growth at this site (Keith et al., 2012), but BIOS-2 does not currently simulate the impact of disturbance including insect attacks. Although the model produces daily carbon and water fluxes, we used monthly rather than daily flux observations because the precipitation data are more reliable at monthly timescales (Jones et al., 2009), as follows: Precipitation is spatially very variable at daily timescales, thus difficult to observe accurately with the relatively sparse gauge network. The spatial pattern of precipitation at monthly timescales is significantly smoother, therefore more accurately interpolated between the measurement locations. The daily precipitation data we use in the model has been rescaled (Jones et al., 2009), so that the sum of the daily values over a month is consistent with the interpolated monthly values.

We also used long-term averaged streamflow at 51 unimpaired catchments (Vaze et al., 2011; Zhang et al., 2011). Monthly mean streamflow was calculated from daily measurements for months with data available for at least 90% of days. The long-term means were then calculated by averaging the monthly means where they exist, for comparison with modelled long-term means (calculated by averaging modelled streamflow for the same months). Long-term means are used for streamflow observations because BIOS-2 does not model streamflow dynamics well, something that we plan to address in future work. Long-term means of leaf NPP, above-ground phytomass, above-ground litter and soil carbon density in the top 15 cm (Barrett, 2001; Raison et al., 2003) were also used. Figure 2 shows the location of the observations used for calibration.

The observations used to optimise CABLE parameters were ET, GPP, streamflow and leaf NPP. The observations used to optimise CASA-CNP parameters were NEP, soil carbon and above-ground phytomass and litter. Haverd et al. (2013b) used many of the same observations, except in that study NEP was not used for parameter optimisation and the eddy flux data were processed in a different way.

### 2.4 Optimisation method

To optimise parameters, we used the PEST implementation (Parameter ESTimation, http://www.pesthomepage.org) of the Gauss-Marquardt-Levenberg method (Doherty, 1999). This is a down-gradient search method, meaning that it uses information about the gradient of the cost function with respect to the parameters to decide how to iteratively alter parameters to locate parameter values corresponding with the minimum in the cost function (Raupach et al., 2005). The cost function that was minimised was the sum of weighted square residuals ($\Phi$) plus the mismatch of some parameters from prior estimates ($\Phi_P$) where we were confident in our prior estimates based on the literature. Flux observations were weighted in $\Phi$ so that each flux site contributed equally to the cost function, regardless of the length of the record used for calibration. The weights for each observation group (e.g. ET, GPP etc) were then scaled so that each group contributed approximately equally to $\Phi$ calculated with the prior parameters. This is important because the different types of observations can have vastly different magnitudes, and the relative contribution of each group to $\Phi$ should not depend on the units that are used. We first optimised CABLE parameters using PEST, then used the optimised CABLE parameters to generate GPP, soil moisture and soil temperature

inputs for CASA-CNP. We then optimised CASA-CNP parameters with PEST. We use the parallel implementation of PEST called BEOPEST.

Due to the large number of processes and parameters, calibration of land surface models is generally an under-determined problem, where there is no unique, correct parameter set and multiple parameter combinations can give an adequate match to observations. Doherty et al. (2010) describes some of the issues involved in optimisation of highly parameterised models. Some combinations of parameters are informed by the calibration dataset, and these are described as comprising the 'calibration solution space'. Errors in solution space parameter combinations are due to measurement noise. Orthogonal to the calibration solution space is the 'calibration null space', containing combinations of parameters that are not informed by the calibration dataset. Expert knowledge, where available, gives the best estimates of parameters that are part of the calibration null space. Often, parameters that have little effect on model outputs for comparison with the calibration dataset are fixed at prior values, this is sometimes called regularisation. As discussed by Doherty (2015), "the purpose of regularisation is to attain uniqueness where none in fact exists". We should avoid simply fixing parameters to possibly incorrect values just because the model outputs that correspond to calibration observations are not sensitive to them. If models are used to make predictions of quantities unlike those used for calibration of parameters, we need to be aware that the predictions may be sensitive to parameters, or parameter combinations, that were not well constrained through the calibration process (Doherty and Johnston, 2003). If this occurs, the uncertainties in these predictions are likely to be under-estimated.

Haverd et al. (2013b) used parameter sensitivity analysis to choose which parameters to optimise, avoiding parameters that were unlikely to be constrained by the available observations. Here we optimise a larger number of parameters, including some of the parameters that are not well constrained by the calibration dataset, to explore the effect of different values of the poorly-constrained parameters on model predictions. Instead of aiming to estimate a single parameter set, we generate an ensemble of parameter sets, to represent parameters that are not well constrained. We can then use this ensemble of parameter sets to see the effect of uncertainty due to parameter equifinality in model predictions, similar to Chen et al. (2011). We use some of PEST's linear analysis tools, including null space Monte Carlo (NSMC). Null space Monte Carlo (Tonkin and Doherty, 2009) is an efficient way to generate multiple parameter sets that are consistent with the observations. As described in Doherty et al. (2010) and Sepúlveda and Doherty (2015), the NSMC method consists of the following steps: (i) Identify the null-space of the model's parameter field from sensitivities calculated during optimisation. (ii) Generate many stochastic realizations of model parameters. (iii) Project these realizations onto the null and solution spaces. (iv) Retain the null space component, but replace the solution space component with that of the calibrated model. (v) Recalibrate these stochastic parameter sets (only a few iterations). Recalibration is required due to non-linearities in the model and an indistinct boundary between the solution- and null-spaces. (vi) Eliminate any parameter sets that are not considered plausible. Utilities exist as part of the PEST package to perform these steps after initial optimisation of parameters (specifically, we used routines SUPCALC to calculate the dimension of the solution space, RANDPAR to generate random parameter sets, PNULPAR to retain the null-space components of the random parameter sets and replace the solution space components with that of the calibrated model, and PARREP to replace parameters in the model's control file with the new parameter sets in preparation for recalibration by PEST).

The ensemble from the NSMC analysis will include the effect of uncertainty in parameters to which model outputs for comparison with observations are not sensitive, as well as parameters to which the model outputs are sensitive but which are correlated with other parameters, as both of these are part of the calibration null space. In addition, the recalibration process and the fact that solutions with a range of values of $\Phi$ are retained means that the ensemble also accounts for the uncertainty in parameters that are well constrained by observations but affected by measurement noise or model structural error (Sepúlveda and Doherty, 2015).

We generated ensembles of parameter sets in the following way. After optimising CABLE parameters with PEST, we used the NSMC method to generate 30 additional parameter sets for CABLE. Based on their values of $\Phi$, we chose the best 20 of these CABLE parameter sets from the NSMC and used them to generate GPP, soil moisture and soil temperature inputs for CASA-CNP, then we optimised the CASA-CNP parameters with PEST for each of these inputs. This gave us 20 corresponding parameter sets for both models, and we call this ensemble of parameters for both models the 'CABLE parameter ensemble' because they originated from null space Monte Carlo applied to CABLE. We then used the null space Monte Carlo method to generate 30 parameter sets for CASA-CNP, all using a single set of CASA-CNP inputs calculated with parameters from the original optimisation of CABLE. We retained the best 20 of these parameter sets; we call this the 'CASA-CNP parameter ensemble' because it originated from null space Monte Carlo applied to CASA-CNP. Overall, this gave us 40 combinations of CABLE and CASA-CNP parameters. The NSMC method does not specifically calculate the posterior parameter probability distributions, however it is an efficient way to generate multiple parameter sets that span a significant amount of the uncertainty due to equifinality in both models.

Down-gradient methods such as the Gauss-Marquardt-Levenberg method have the important advantage of being much more computationally efficient than global search methods. However, they can suffer from the disadvantage of finding only a local minimum, and not the global minimum (Raupach et al., 2005; Wang et al., 2009). There are a number of ways to reduce the chances of getting stuck in a local minimum with PEST (Doherty, 1999). One way is by the choice of the parameter increments in the different stages of the optimisation (large increments at first, then smaller increments), and how the derivatives are calculated. Another way is by the choice of the initial parameter values (as good as possible, such as based on expert knowledge) or by repeating the optimisation with different initial parameter values. The use of the NSMC method allows a much more thorough search of parameter space than a single PEST optimisation, but in a computationally efficient way. Skahill and Doherty (2006) describe additional ways to significantly improve the chances of finding the global minimum with PEST. Here, we reduce the chances of PEST getting stuck in a local minimum by basing our choices for parameter increments and the derivative calculation method on recommendations in Doherty (1999), and by using the NSMC method.

## 3 Results

In this section, we will focus first on the results of the parameter optimisation (e.g. how well the model matches the observations, which observations inform which parameters, and parameter equifinality), then look at modelled IAV in carbon fluxes

(what the model predicts for Australia, and how parameter equifinality affects our estimates) and our modelled 2011 NEP anomaly.

## 3.1 Optimisation results

### 3.1.1 Comparison of model outputs with observations

Figures 3 and 4 show monthly and annual timeseries of GPP, ET, ecosystem respiration and the anomaly in NEP at three contrasting OzFlux sites: Howard Springs (tropical savanna), Tumbarumba (cool temperate) and Alice Springs Mulga (sparsely vegetated). Timeseries for all 14 OzFlux sites considered here are shown in Supplementary Material (Figs. S2-S9). Our best case corresponds to the case that has the lowest total $\Phi$ (where total $\Phi$ is the sum of $\Phi$ for both CABLE and CASA-CNP calculated separately, i.e. $\Phi_{CABLE} + \Phi_{CASA}$). This also happens to be the case with the lowest $\Phi_{CASA}$. The annual timeseries plots also show model results for the ensembles of parameter sets. In addition, we include one case of ecosystem respiration and NEP calculated with the Kelly et al. (2000) soil respiration function (parameters other than those used in Eq. 1 have been re-optimised for this case). When anomalies are shown for ensemble members here and in subsequent figures, they are calculated for each ensemble member by subtracting the temporal average of the quantity for that ensemble member.

Figure 5 shows results for monthly and annual ET, GPP and ecosystem respiration for the best case plotted as scatter plots of model versus observations, with different colors used for each site. Figure 6 shows the modelled versus measured soil carbon density in the top 15 cm for the best case. Modelled versus measured streamflow, leaf NPP, above-ground litter and above-ground phytomass for the best case are shown in Supplementary Fig. S10. Supplementary Table S1 compares our flux and pool estimates averaged over 1990-2010 for the best case and ensemble mean with values from BIOS-2 in Haverd et al. (2013b). We also show our uncertainty range due to parameter equifinality ($1\sigma$) calculated from the ensemble and the total uncertainty from Haverd et al. (2013b) due to parameter and forcing uncertainty.

There was no apparent relationship between $\Phi_{CABLE}$ and $\Phi_{CASA}$, indicating that a better fit to the observations in CABLE did not lead to a better fit to CASA-CNP observations. Monthly ET, GPP and ecosystem respiration at Howard Springs and Tumbarumba are both dominated by a strong seasonal cycle, whereas the variability at Alice Springs Mulga is very episodic. NEP is the difference between two large fluxes of opposite sign, and it is difficult to model the seasonal cycle well, particularly at Howard Springs where GPP and ecosystem respiration are highly correlated at monthly time-scales.

$\Phi$ for the best case divided by $\Phi$ for prior parameters, split into observation groups, is as follows: ET 0.88, GPP 0.46, NPP 0.08 and streamflow 1.06 for CABLE observations, and NEP 0.32, soil carbon 0.80, phytomass 0.95 and litter 0.78 for CASA-CNP observations. The best to prior ratio for total $\Phi$ was 0.36. Note that the prior CASA-CNP case used prior CASA-CNP parameters but inputs from CABLE calculated with optimised CABLE parameters, and therefore shows the change in the agreement with observations due to optimisation of only CASA-CNP parameters. Optimisation of parameters has improved the agreement with many of the observations, but has degraded the fit to a few observations. For example, run-averaged NPP is significantly improved by parameter optimisation, but run-averaged streamflow is slightly worse. Figure 4d shows that mean GPP at Howard Springs has moved away from the observations (the parameter responsible for this change is vcmax_g, the

maximum RuBP carboxylation rate to leaf for grass, that has moved away from its prior value). The degradation of the fit to some observations is a consequence of trying to fit many different types of observations at once with a complex model, and it is not entirely surprising that there are some discrepancies. Richardson et al. (2010) pointed out that this often occurs. Nonetheless, it is an indication of deficiencies in the model, including the forcing and specification of parameters, and/or the observations and their uncertainty characterisation, but we have not yet been able to identify the specific causes of these deficiencies in our model.

Overall we capture the observed level of NEP variability well. The agreement with observed annual NEP flux anomalies at the OzFlux sites has improved relative to the original BIOS-2 calculations: RMSE 0.39 $gCm^{-2}d^{-1}$ (this work) compared with 0.58 $gCm^{-2}d^{-1}$ (Haverd et al., 2013a), an improvement that is possibly attributable to optimization of the heterotrophic respiration response to soil moisture. However, the correlation between modelled and observed annual NEP values is poor ($R^2 = 0.1$, Fig. 5f). IAV in NEP at flux sites is difficult to capture well in the model for a number of reasons: 1) NEP is the difference between two very large fluxes, and these fluxes are temporally highly correlated in the non-temperate regions, with both fluxes being highly sensitive to soil moisture. 2) Flux measurements are quite local, whereas the model has a resolution of $0.05^o x 0.05^o$. 3) We are missing some processes from the model, such as disturbance (e.g. the insect attack at Tumbarumba) and fire that may be important at the local scale of the flux measurements. 4) Flux measurements are also subject to errors, particularly due to the partitioning algorithm.

### 3.1.2  Observation worth

Fig. 7 shows how each observation group contributes to the reduction of uncertainty in CABLE parameters. In Fig. 7a we show how the post-calibration uncertainty variance *increases* as each observation group is left out one at a time. A rise in uncertainty variance occurs for observation groups that contain unique information about a parameter that is not contained in the other groups. Figure 7b shows the *decrease* in pre-calibration uncertainty variance for each observation group used on its own. A decrease will occur when any observation group contains information about a parameter, even if this information is redundant. For example, the parameter lgamma_g (that controls the drought response of grass stomatal conductance) is constrained by all observation groups, and leaving out any individual observation group makes little difference to the post-calibration uncertainty, indicating redundancy in the information provided by the observation groups. Many of the CABLE parameters are constrained by more than one observation group. In contrast, parameters alloclg and alloclw (describing allocation of carbon to leaves) are mainly constrained by NPP observations, and leaving NPP observations out of the optimisation significantly increases the uncertainty in these parameters. The eddy flux data (ET and GPP) provide the tightest constraints on the biophysical parameters, as also found by Haverd et al (2013a), presumably because they contain temporal information. Streamflow seems to contain mostly redundant information that is available from the other observations, but is still worth including to mitigate against the effect of biases in any single observation type. In future work, we plan to improve streamflow dynamics in the model, and would then hope to take advantage of temporal information in the streamflow measurements.

The value of observation groups to estimation of CASA-CNP parameters is shown in Supplementary Fig. S11. Many of the CASA-CNP parameters are not well constrained by the calibration observations. Most of those that are constrained to some

extent are influenced by only one observation group, demonstrating the benefit of including all four observation groups. Specifically, parameters age_leaf_w and age_clitt2 describing turnover times of woody leaves and fine structural litter are influenced by observations of above-ground litter; age_wood describing turnover of wood and falloc_w describing the fraction of carbon allocated to wood are influenced by measurements of above-ground phytomass; and parameter soilc0_frac for the fraction of soil carbon in the top 15 cm and soil carbon pool turnover times age_csoil1, age_csoil2 and age_csoil3 are influenced by observations of soil carbon. The function describing the effect of soil moisture on soil respiration is constrained by observations of both NEP and soil carbon. This analysis of which observation groups constrain which parameters for both CABLE and CASA-CNP gives results that are as we would have expected. The observation worth is calculated using PEST's linear analysis tools (routine GENLINPRED).

### 3.1.3 Parameter equifinality

Figure 8 shows scatter plots of the model-data mismatch for CABLE observations, $\Phi_{\text{CABLE}}$, against each CABLE parameter, where the range of the x-axis corresponds to the prior range for the parameter. For reference, $\Phi_{\text{CABLE}}$ with prior parameters was 3962. Relative to their prior range, some parameters cover a small range for low values of $\Phi_{\text{CABLE}}$, such as alloclw, alloclg (leaf carbon allocation coefficients in leaves and grass) and lgamma_g (controls drought response of grass stomatal conductance), indicating that they are relatively well constrained by the optimisation. Other parameters, such as f10_w (fraction of woody roots in the top 10cm), dleaf_g (grassy leaf length) and zr_g (maximum grassy rooting depth) cover a wide range of parameter values (relative to their prior range) for very little variation in $\Phi_{\text{CABLE}}$, implying that their value is not so well constrained by the optimisation. In some cases, combinations of parameters might be well constrained by the optimisation but the individual parameters are not. We can see this by looking at the parameter identifiability (Doherty and Hunt, 2009), based on analysis of the posterior parameter covariance matrix using tools that are available with PEST (routine IDENTPAR). Figure 9 shows the identifiability of combinations of parameters in CABLE. Early eigenvectors (dark colours) are most identifiable (comprise the calibration solution space), later eigenvectors (pastel colours) are least identifiable (comprise the calibration null space). Eigenvectors split across parameters indicate whether combinations of parameters, rather than individual parameters, are identifiable. In general, parameters with the smallest ranges for low $\Phi_{\text{CABLE}}$ in Fig. 8 are part of the most identifiable eigenvectors, as expected. The parameter f10_w (fraction of grass roots in the top 10cm) that had a wide range for low values of $\Phi_{\text{CABLE}}$ in the scatter plots in Fig. 8, has identifiability that is comprised of a number of the most identifiable eigenvectors, as indicated by the dark colors. Whether it is a problem that parameters are not individually identified depends on what the model is being used to predict or calculate.

Scatter plots and parameter identifiability for CASA-CNP are shown in the Supplementary Figs. S12 and S13. $\Phi_{\text{CASA}}$ for prior parameters (but using inputs from CABLE calculated with optimised parameters) was 1449. Many of the CASA-CNP parameters are not well constrained relative to their prior ranges by the observations. The soil respiration function seems to be fairly well constrained compared to the range of curves shown in Fig. 4c of Sierra et al. (2015), and parameters s and $w_1$ in this function are among some of the best constrained parameters. Parameters $w_2$ and $w_3$ are unconstrained, but due to the values of the other parameters in this function they are not used they so are irrelevant to the model.

### 3.1.4 Model structural choices

Without the shade correction (Eq. 5), the agreement with calibration observations is a bit worse than our best case for some observation types (e.g. the ratio of optimised to prior $\Phi$ for the noshade case for NPP and soil carbon are 0.27 and 0.91, compared to 0.08 and 0.80 for our best case) and a bit better for others (GPP and NEP $\Phi$ ratio for the noshade case are 0.32 and 0.29, compared to 0.46 and 0.32 for our best case), but overall the total $\Phi$ is not significantly different. We have used the shade correction here because it is more physically realistic, although the comparison with observations does not favour either parameterisation.

The function describing the effect of soil moisture on soil respiration for our ensemble of parameter sets is shown in Fig. 1b, with lines colored by the total $\Phi$ from both models. The optimised functions are all quite different to the function from Kelly et al. (2000), with the optimised functions increasing throughout the range of $\bar{s}$ from 0 to 1.0, rather than having a peak followed by a decrease. A case using the Kelly et al. (2000) function with the other CASA-CNP parameters re-optimised has a higher value of $\Phi$, particularly for NEP observations, but this is to be expected when we optimise fewer parameters.

### 3.2 Interannual variation in NEP for Australia

Figure 10 shows modelled annual values of NEP anomaly for six bioclimatic regions and the continent. Results are shown for both the CABLE and CASA-CNP ensembles of parameters in grey. The red lines show the case re-optimised without the shade correction for deriving vegetation cover from fPAR, and the blue lines show the case re-optimised with the Kelly et al. soil respiration dependence on soil moisture. The bioclimatic regions, shown in Fig. 10h, are an aggregation of the agro-climatic classification of Hutchinson et al. (2005) into six classes, as described and used by Haverd et al. (2013a, b). Annual values of GPP and heterotrophic respiration anomaly for the bioclimatic regions and Australia are shown in Supplementary Figs. S14 and S15. Supplementary Fig. S14 shows total NPP as well as the contributions from grassy and woody vegetation. IAV in NPP for grassy vegetation types is larger than the IAV in NPP for woody types in the tropics, savanna, Mediterranean and Australia. IAV in NPP for woody vegetation is similar to or larger than IAV in NPP for grassy vegetation in warm and cool temperate regions.

Figure 11 shows the modelled annual NEP anomaly together with annual precipitation for each region and the continent, and the Southern Oscillation Index. There is a strong relationship between NEP and precipitation in all regions, as has been shown in many previous studies. Precipitation is clearly the most important factor influencing interannual variations in NEP, predominantly precipitation in the current year but also to some extent precipitation in the years leading up to the current year due to 'memory' effects (Schimel et al., 2005; Poulter et al., 2014). Our results show high NEP anomalies for Australia in 1983-84 (after a strong El Nino), 1989 (after a strong El Nino), 2000 (in the middle of a prolonged La Nina but Australian precipitation was very high) and very high values in 2010 and particularly 2011 (precipitation was very high, at the end of a decade-long drought). We see low NEP anomalies in 1982 (during a strong El Nino), 1994 (during the third of three consecutive El Ninos, with very low precipitation), 2002 (at the beginning of an El Nino, with very low precipitation) and 2014 (at the beginning of an El Nino). Using an earlier version of the BIOS-2.1 configuration (but without the correction to

the vegetation cover for shaded grass), Haverd et al. (2016b) found no significant change in the sensitivity of Australian NEP to rainfall, contrary to the suggestion by Poulter et al. (2014) of a shift during recent decades in the sensitivity of vegetation activity to moisture availability. Haverd et al. (2016c) also used the BIOS-2.1 model and showed that at continental scale, annual variations in production are dampened by annual variations in decomposition, with both fluxes responding positively to

precipitation anomalies, in contrast to previous global modelling results (Poulter et al., 2014) suggesting that IAV in Australian net carbon uptake is amplified by lags between production and decomposition.

Some years have significant range in NEP anomaly due to parameter equifinality, usually when the anomaly is furthest from zero (either positive or negative), while at other times the range is quite small. The range is particularly large compared to the calculated IAV in the tropics, medium in the warm and cool temperate and fairly small in the other regions and Australia as a

whole. The range in heterotrophic respiration is larger than the range in NPP in the tropics and temperate regions, but they are similar in other regions and for Australia as a whole (Figs. S14 and S15).

Annual NEP anomalies for a case optimised without the shade correction are shown by the red lines in Fig. 10 and are mostly within the range given by the ensemble of cases with the shade correction. We generated an ensemble of results for the case without the shade correction (not shown here but used in Haverd et al. (2016b)) and it had a similar spread of results to the

cases shown here. Therefore we place no importance in the difference between the red lines and our other cases. Annual NEP anomalies for a case calculated using the Kelly et al. (2000) soil respiration function are shown by the blue lines in Fig. 10 and fall at the high IAV end of the range given by the other cases.

We estimate that IAV in Australian NEP is 0.12–0.21 PgC yr$^{-1}$ for the period 1982-2013. This quantity is the standard deviation of annual NEP anomalies calculated separately for each ensemble member, with the range given by the ensemble.

NEP IAV relative to mean NPP is 6–10%. In the earlier BIOS-2 implementation, Haverd et al. (2013b) gave a continental value of 8% for NEP variability ($1\sigma$) relative to mean NPP for the period 1990–2011. Over this shorter period, our range is 7–10%.

### 3.3  2011 NEP anomaly

Our estimate for the 2011 NEP anomaly (relative to the 1982-2013 mean) is 0.43–0.67 PgC yr$^{-1}$. In Figure 12a we show the ensemble of estimates of the 2011 anomaly plotted against the corresponding model-data mismatch (total $\Phi$). There seems to

be no relationship between the modelled magnitude of the 2011 peak and how well the model fits the calibration observations, as we see quite different values of 2011 NEP for very similar values of total $\Phi$. Figure 12b shows the size of the 2011 anomaly against $\Phi$ for just the monthly NEP flux measurements ($\Phi_{NEP}$). While there is a suggestion of a relationship here (lower $\Phi_{NEP}$ corresponds to lower 2011 anomaly values), we note that the flux measurements are not without error, NEP at the flux sites is difficult to model well and the model does not include all of the processes that may be important at the local scale. We therefore

do not have particularly higher confidence in the estimates that have better agreement with flux observations, but prefer to take account of the agreement of the model to all of the different types of measurements. Previous studies (e.g. Fox et al., 2009; Richardson et al., 2010; Keenan et al., 2012b; Luo et al., 2015; Du et al., 2015) have emphasised the importance of using both pool- and flux-based datasets to constrain land surface models, and a strength of our work is that we have used observations

of both types in this study. In Fig. 12c we show the ensemble of estimates of the 2011 anomaly plotted against IAV ($1\sigma$), indicating a strong relationship between the size of the 2011 anomaly and overall IAV of each ensemble member.

The 2011 NEP anomaly stands out as extreme compared to all other years. The best case (with lowest total $\Phi$) has 2011 NEP anomaly near the lower end of the range (0.47 PgC yr$^{-1}$). However during the development of this work we generated a

few different ensembles of parameter sets with only small differences to the model and inputs, and found that the range stayed quite constant but that the 2011 anomaly for the best case could be anywhere within the range. Parameter equifinality has an important effect on our estimate of the 2011 NEP anomaly, and we are currently not able to identify where within the range 0.43–0.67 PgC yr$^{-1}$ the true value is most likely to sit. Further constraints on the model parameters are needed to reduce the uncertainty in this estimate.

Using the Lund-Potsdam-Jena (LPJ) dynamic global vegetation model, Poulter et al. (2014) estimated a 2011 NEP anomaly relative to the 2003-2012 mean for Australia of 0.66 PgC yr$^{-1}$. Our estimate for the 2011 NEP anomaly relative to the 2003-2012 mean is 0.40-0.61 PgC yr$^{-1}$, just under the estimate from Poulter et al. Like Poulter et al., we see that the IAV in NEP (including the 2011 anomaly) is dominated by IAV in NPP rather than respiration. Our uncertainty in the NEP anomaly has roughly equal contributions from equifinality in parameters important for NPP and heterotrophic respiration for Australia

and the savanna, sparsely vegetated and Mediterranean regions. Uncertainty in NEP in the tropical and temperate regions is dominated by the uncertainty due to parameter equifinality in ecosystem respiration rather than NPP.

## 4   Discussion

Our correlation of modelled and observed annual NEP at the flux sites is quite poor and, despite a reduction in $\Phi_{\text{NEP}}$ by optimisation of parameters, IAV in annual NEP at the flux sites appears not to be significantly improved by optimisation.

Previous studies have found that land surface models have difficulty simulating the correct timing of IAV in carbon fluxes (e.g. Urbanski et al., 2007; Keenan et al., 2012a, b), although see Desai (2010). A key point is that IAV in NEP at the flux sites is not particularly representative of IAV in NEP for the country as a whole. Other than Alice Springs, measurements at the flux sites do not show a strong relationship between NEP and available soil water. This is in contrast to the parts of the country that most influence the continental NEP, where vegetation growth is mostly water limited. Using BIOS-2.1, Haverd et al. (2016c)

found that 90% of Australian IAV in NEP is due to the savanna and sparsely vegetated regions, and the majority of the flux sites are outside these regions.

Without many flux observations at sites that are water limited (the record from Alice Springs is currently only a few years long), it is difficult for us to assess how well the model simulates regional and continental carbon fluxes for Australia. Additional flux observations at sites that are water limited, and therefore more representative of Australian carbon fluxes, might help us to

assess how well the model matches observations, and would also be valuable for parameter estimation and model development. It is not clear whether the meteorological drivers can explain the IAV at the current flux sites, and a study similar to Abramowitz et al. (2008) using statistical models but focused on the interannual timescale at Australian sites may be useful to answer that question.

Differences in the present study from Haverd et al. (2013a) include the use of an improved product for vegetation cover (GIMMS NDVI3g) that extends over several decades, the correction for shaded grass used in calculating the vegetation cover, the use of OzFlux NEP observations for parameter optimisation, inclusion of the function describing the effect of soil moisture on soil respiration in the optimisation (Haverd et al. (2013a) manually tuned this function), optimisation of a greater number of parameters, and more rigorous analysis of parameter uncertainty by the generation of multiple parameter sets that are used to explore parameter equifinality. Our estimates for the carbon pools and fluxes generally agree with Haverd et al. (2013a) within the uncertainty ranges (Supplementary Table S1). An exception is the fraction of continental NPP attributable to recurrent (assumed grassy) vegetation, which is $0.40 \pm 0.04$ ($1\sigma$), compared with $0.67 \pm 0.14$ in the 2013 analysis. Litter pools are also higher: continental average of $8.4 \pm 2.3$ tCha$^{-1}$, compared with $2.5 \pm 1.3$ tCha$^{-1}$ in the BIOS-2 analysis. The increase in litter in the current work is attributable to a correction to the Haverd et al. (2013a) analysis in which litter observations were incorrectly assumed to be comparable with total (above and below-ground) fine-structural litter, when in fact they should be compared with only the above-ground component. We now have increased confidence in our estimates for IAV, principally due to the use of the improved product for vegetation cover, optimisation of the soil respiration function and more rigorous parameter uncertainty analysis.

We have focused here on the range of model results that come from parameter equifinality when many other choices are fixed, such as model structure, choice of the cost function and weights for observations and prior estimates of parameters, observations and forcing data. Many of these choices can also lead to uncertainty in the results. We have not calculated the total uncertainty here. Our range of results due to equifinality highlights the dangers of taking a single parameter set. In particular, comparison of model results for the single best parameter set for two different model configurations could easily lead to incorrect conclusions if the effect of parameter equifinality was ignored. Different types of observations can be tested to see whether they would reduce the uncertainty in parameters that are not well constrained. Future work already underway will include the effect of nutrients, land-use change, fire and tree demography on Australian carbon fluxes, with a more comprehensive assessment of the uncertainties.

## 5  Conclusions

We have used a multiple constraints approach to optimise model parameters in BIOS-2.1, an updated fine resolution implementation of the CABLE, CASA-CNP and SLI models for Australia, with a particular focus on interannual variability. In addition to other parameters, we optimised a function for the dependence of soil respiration on soil moisture. We have explored the effect of parameter equifinality on calculated interannual variation in NEP anomalies. The timing of interannual variations in NEP at the flux sites is not particularly well captured by the model, as has been found in previous modelling studies, however most of the flux measurements are from locations that are not water limited, in contrast to the parts of the country that most influence Australian NEP. We estimate that the $1\sigma$ variation in IAV in Australian NEP is 0.12–0.21 PgC yr$^{-1}$. The value of the IAV in NEP is dominated by NPP, but the range of estimates due to parameter equifinality has roughly equal contributions from parameters associated with both heterotrophic respiration and NPP. The 2011 Australian NEP anomaly relative to the

1982-2013 mean is 0.43-0.67 PgC yr$^{-1}$. We find a strong relationship between the size of the 2011 anomaly and the overall IAV. Our range of results due to parameter equifinality demonstrates how errors can be underestimated when a single parameter set is used.

*Acknowledgements.* This work has been undertaken as part of the Australian Climate Change Science Program, funded jointly by the Department of the Environment, the Bureau of Meteorology and CSIRO. We thank researchers in the Australian Ozflux network for making the OzFlux data available. We thank Ranga Myneni and Jian Bi for supplying GIMMS NDVI3g updated to 2013, and Randall Donohue for processing these data. We thank John Doherty of Watermark Numerical Computing for making PEST freely available, and for advice on the use of PEST and its utilities. Tilo Ziehn provided helpful comments on the manuscript.

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

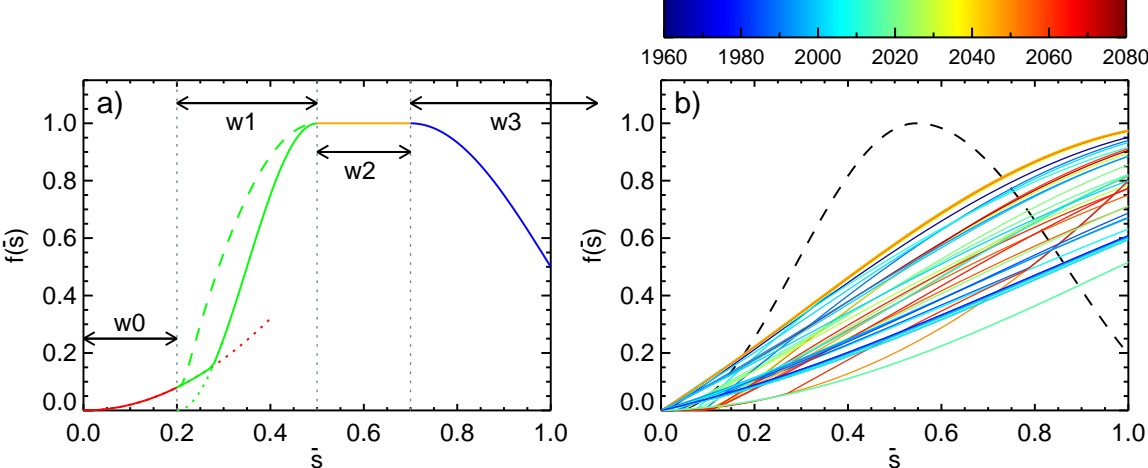

**Figure 1.** Function describing the heterotrophic respiration dependence on soil moisture, $\bar{s}$. a) Schematic figure to explain the form and parameters in the function in Eq. 2. Parameters used for the example shown by the solid line were $(q, c, w_0, w_1, w_2, w_3) = (0.3, 0.2, 0.6, 1.0, 0.2, 2.0)$. b) The dashed black line shows the function given in Kelly et al. (2000) for heterotrophic respiration dependence on soil moisture. The solid lines show our estimated function for the ensemble of parameters, colored by the corresponding value of $\Phi$ for both CABLE and CASA-CNP combined.

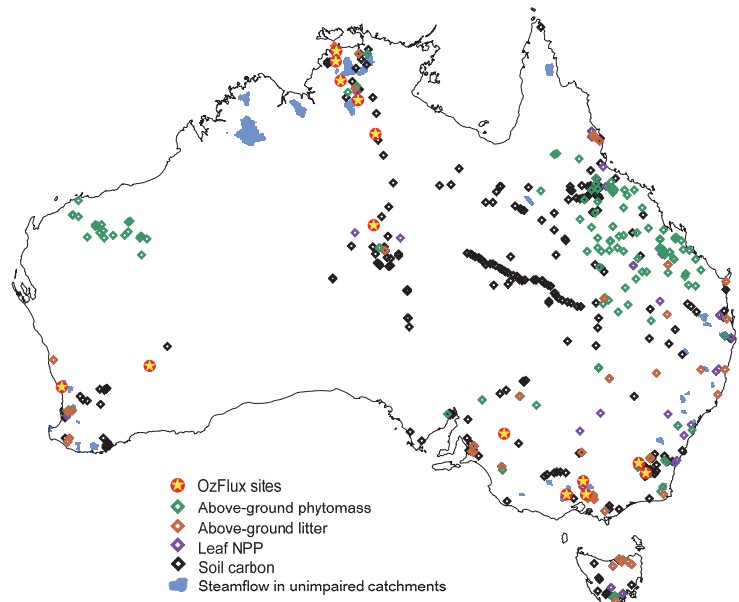

**Figure 2.** Location of observations used for calibration. All OzFlux sites used in this study are shown (i.e. those used for both calibration and validation).

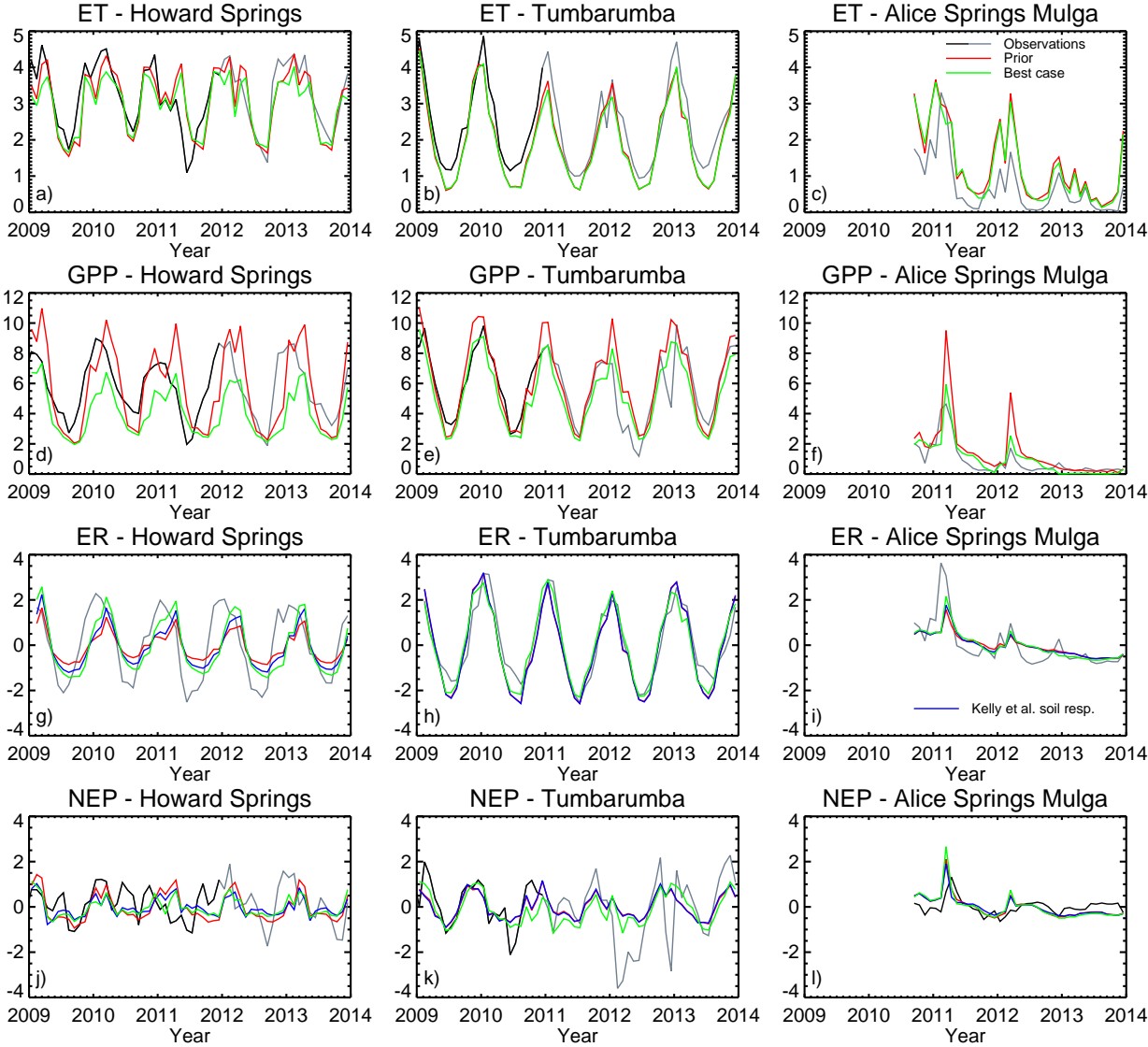

**Figure 3.** Monthly timeseries of ET (mm d$^{-1}$), GPP (gC m$^{-2}$ d$^{-1}$), ecosystem respiration (ER, gC m$^{-2}$ d$^{-1}$) and NEP (gC m$^{-2}$ d$^{-1}$) at Ozflux sites Howard Springs, Tumbarumba and Alice Springs Mulga. Black lines show observations used for calibration, grey lines show observations left for validation. Green lines show modelled quantities for optimised parameters corresponding to the lowest combined $\Phi$ for both CABLE and CASA-CNP. Blue lines (for ecosystem respiration and NEP) show the case using the Kelly et al. (2000) soil respiration function. Red lines show model quantities corresponding to prior parameters. Note that the ER and NEP from CASA-CNP calculated with prior parameters have used inputs from CABLE with optimised parameters, to indicate the effect of optimisation of CASA-CNP parameters only.

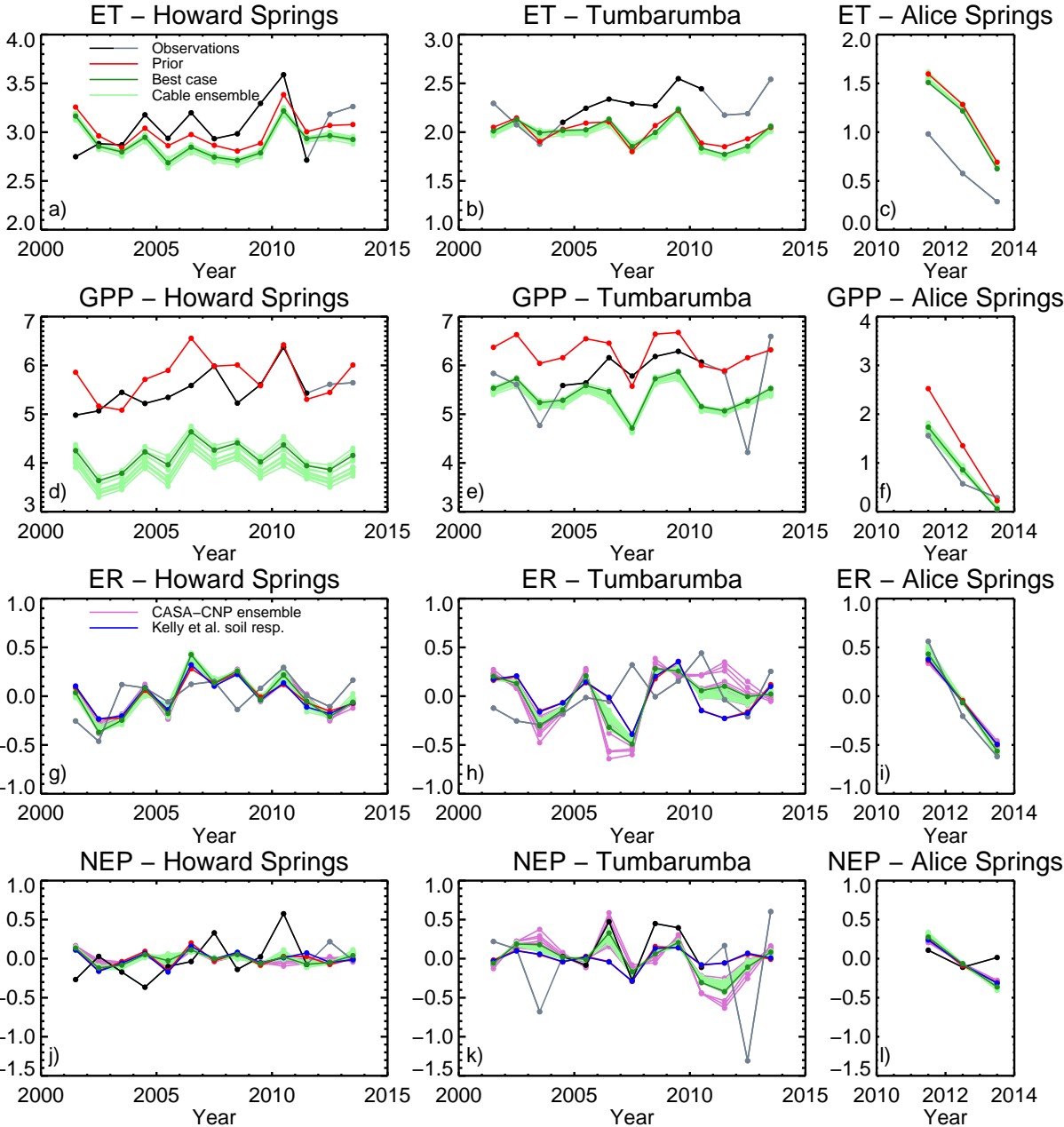

**Figure 4.** Annual mean timeseries of ET (mm d$^{-1}$), GPP (gC m$^{-2}$ d$^{-1}$), ecosystem respiration (ER, gC m$^{-2}$ d$^{-1}$) and NEP (gC m$^{-2}$ d$^{-1}$) at Ozflux sites Howard Springs, Tumbarumba and Alice Springs Mulga. Line colors are as in Figure 3 with the addition of light green lines for the CABLE parameter ensemble and pink lines for the CASA-CNP parameter ensemble.

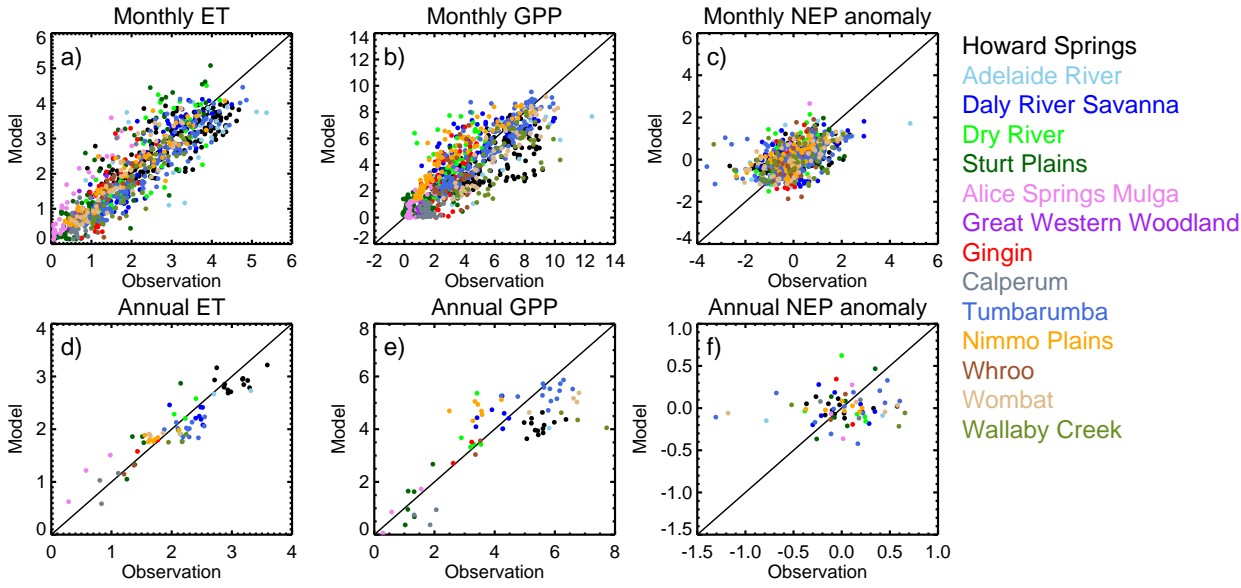

**Figure 5.** Scatter plots of modelled versus observed (best case) monthly and annual ET (mm d$^{-1}$), GPP (gC m$^{-2}$ d$^{-1}$) and NEP (gC m$^{-2}$ d$^{-1}$) at 14 Ozflux sites. Symbols are color-coded according to site.

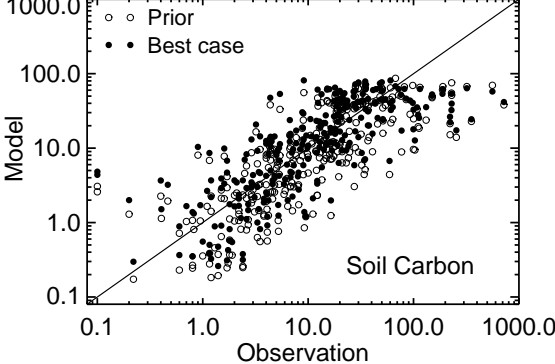

**Figure 6.** Scatter plot of modelled versus observed (prior and best case) long-term averaged soil carbon density in the top 15 cm.

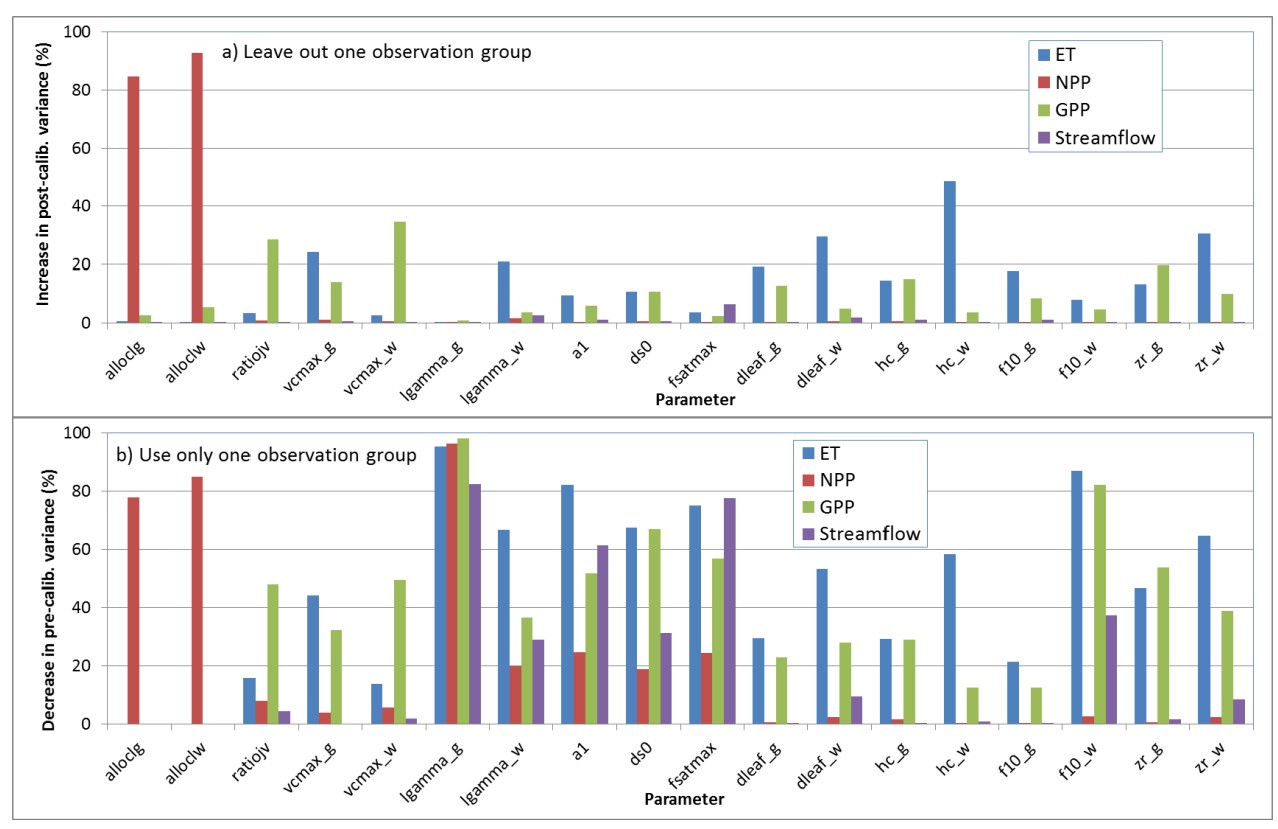

**Figure 7.** Worth of the different observation groups to the estimate of each CABLE parameter. a) Increase (%) in post-calibration parameter uncertainty variance incurred through loss of observation groups (i.e. leaving out each group in turn). b) Decrease (%) in pre-calibration parameter uncertainty variance incurred through addition of observation groups (i.e. each observation group is used on its own).

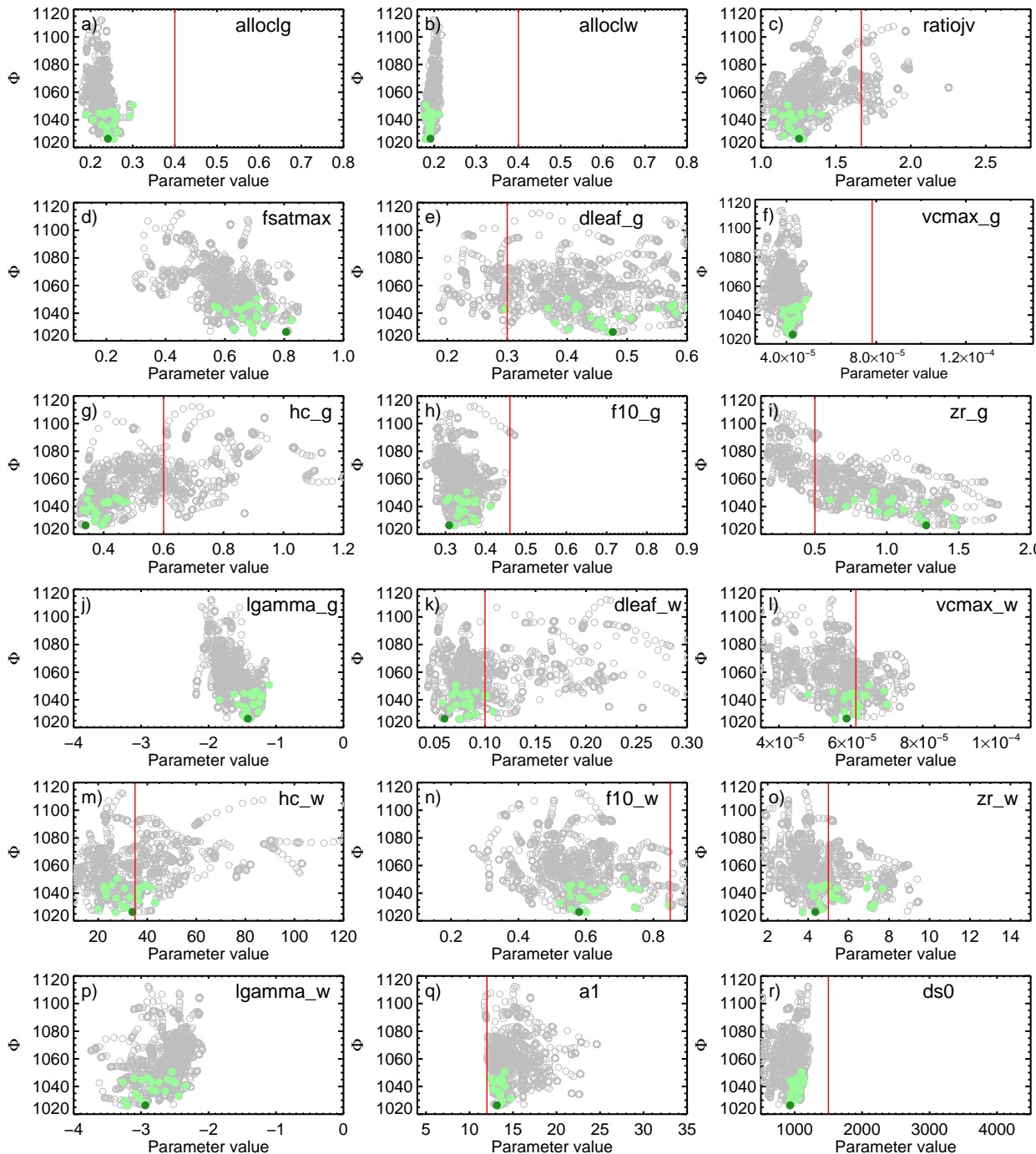

**Figure 8.** Model-data mismatch for CABLE observations, $\Phi_{CABLE}$, plotted against parameter values for CABLE. Grey symbols are all parameter sets tested during the null space recalibration. Light green symbols show the CABLE parameter ensemble plus the original CABLE optimisation. The dark green symbol shows the parameter set that gives the lowest combined $\Phi$ for both models. Red vertical lines show the prior parameter constraints (not all parameters had prior constraints).

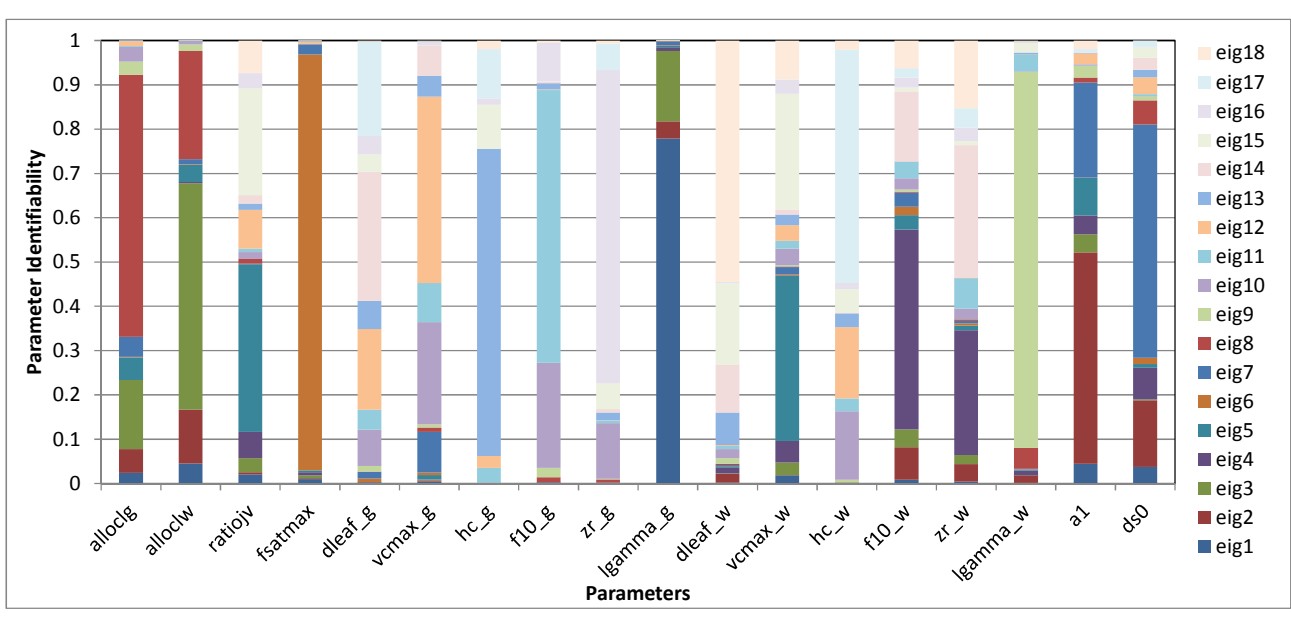

**Figure 9.** Parameter identifiability for CABLE parameters from PEST's linear analysis tools. Dark colors indicate eigenvectors that are more identifiable than light colors.

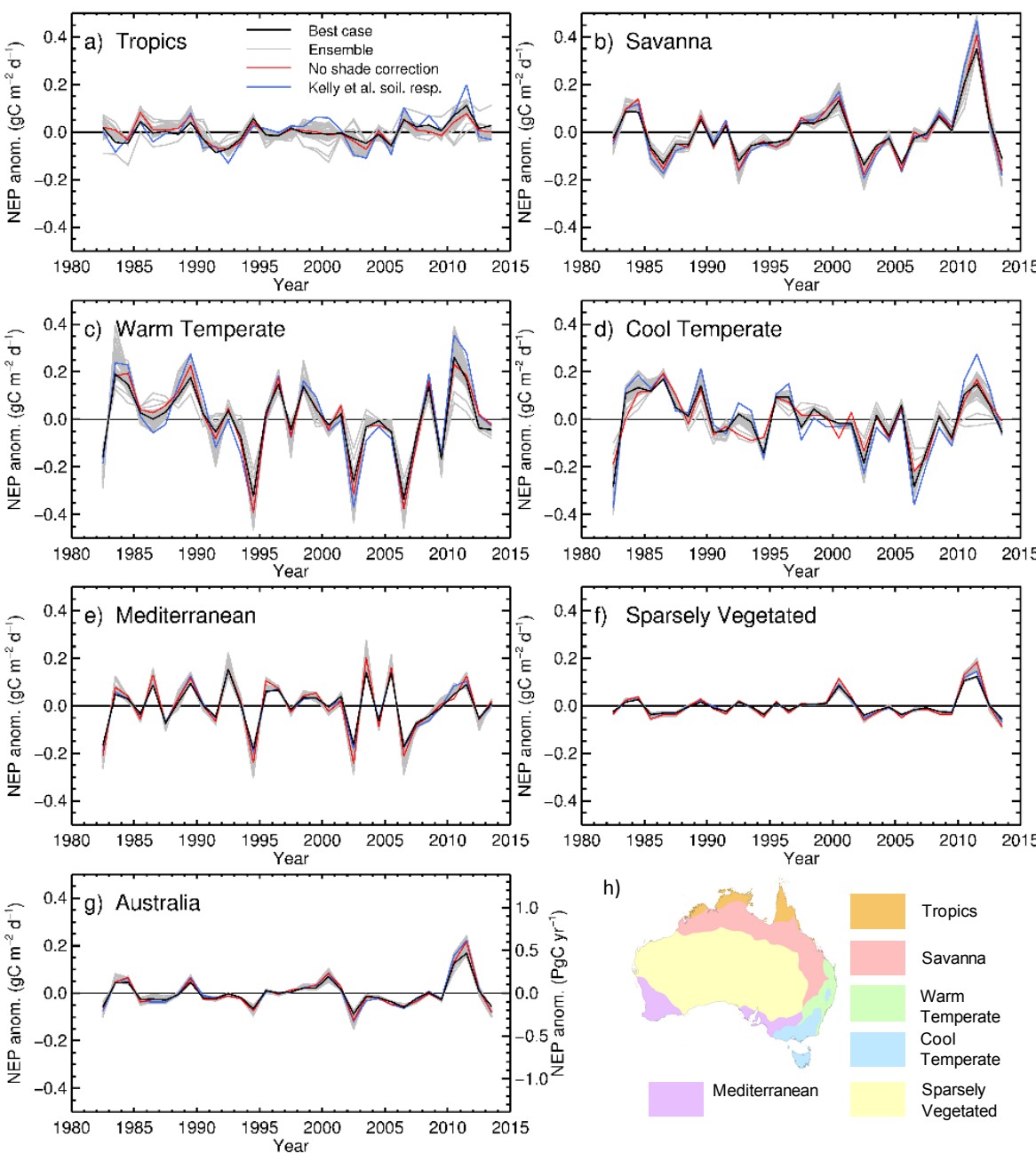

**Figure 10.** Annual anomalies in net ecosystem production for six bioclimatic regions and Australia. The best case is shown in black, with the other ensemble members in grey to indicate the influence of parameter equifinality. The red line corresponds to the case re-optimised without the correction to the vegetation cover for shaded grass. The blue line corresponds to the case re-optimised with the Kelly et al. (2000) soil respiration function. The y-axis range is the same in all panels. Units are gC m$^{-2}$ d$^{-1}$, but panel g) also shows units of PgC yr$^{-1}$ on the right. Panel h) shows the location of the bioclimatic regions.

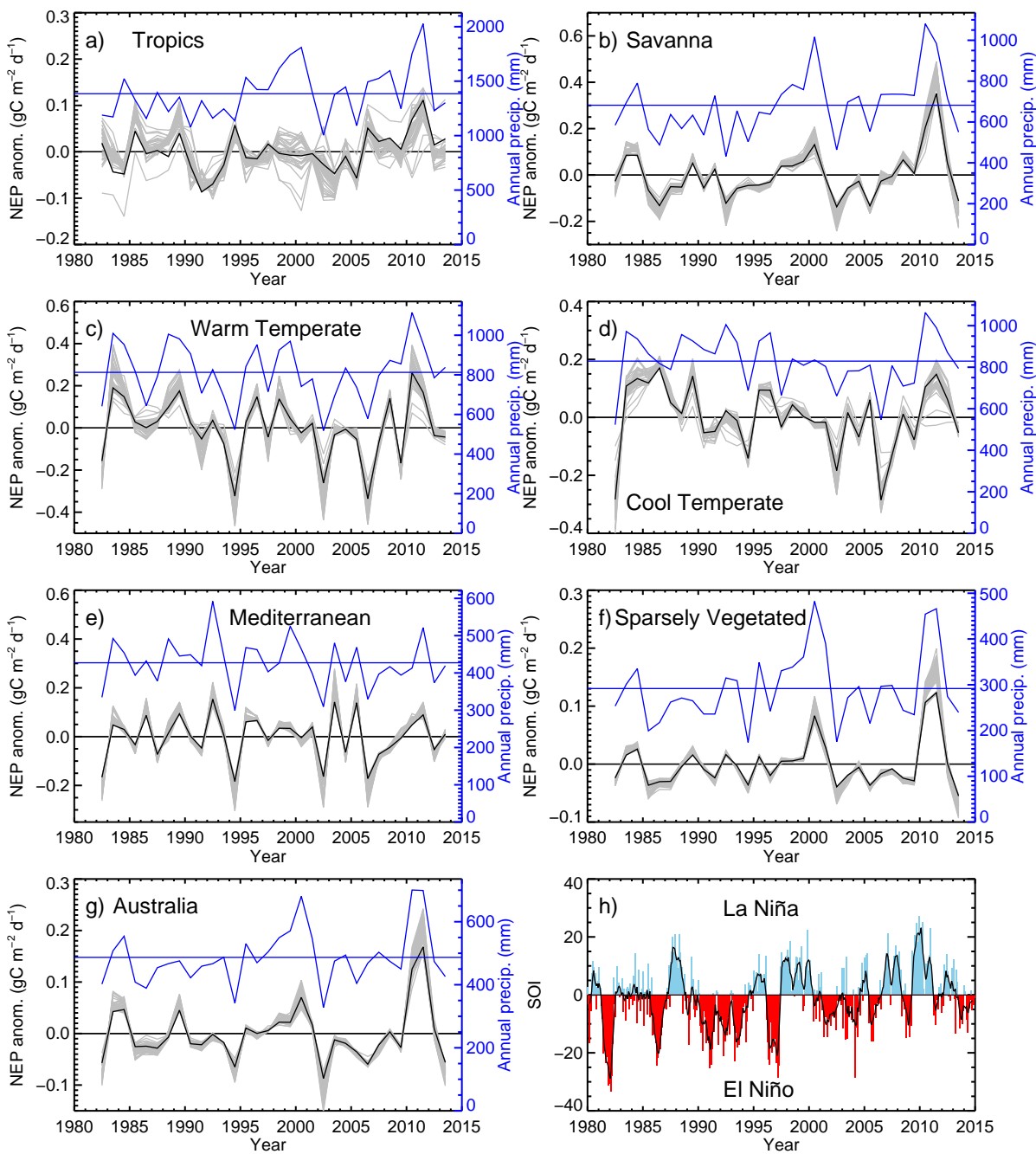

**Figure 11.** a) to g) Annual anomalies in net ecosystem production for six bioclimatic regions and Australia. Black lines show the case with lowest Φ and the grey band is the range due to parameter equifinality excluding the two outlier cases. Blue lines show annual precipitation (mm) for each region. Note that the y-axis range is different for each region. h) Southern Oscillation Index from http://www.bom.gov.au/climate/current/soihtm1.shtml.

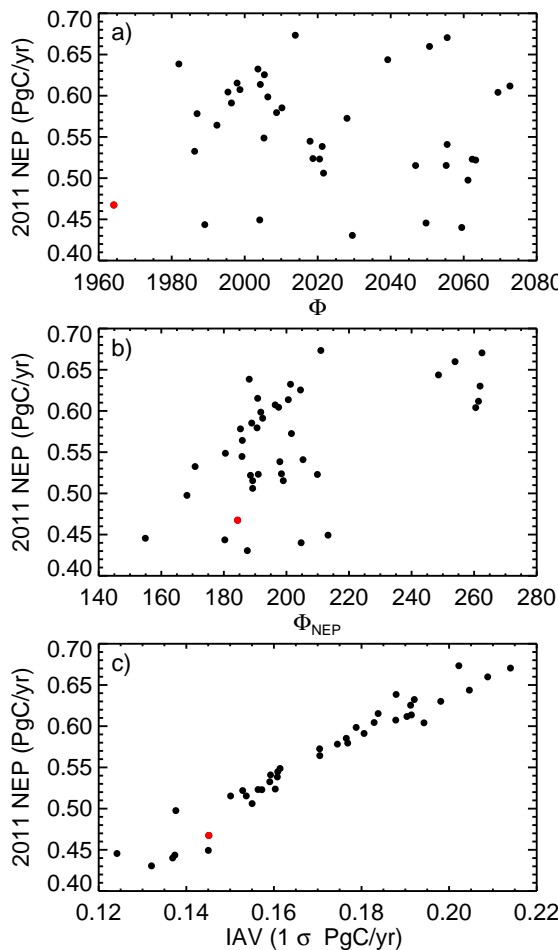

**Figure 12.** a) Australian NEP anomaly in 2011 relative to 1982-2013 for the ensemble of estimates plotted against the value of the total $\Phi$ from both models. b) Australian 2011 NEP anomaly plotted against the value of $\Phi$ from just the monthly NEP flux observations ($\Phi_{NEP}$). c) Australian 2011 NEP anomaly plotted against the IAV for Australia over 1982-2013, expressed as $1\sigma$. In all panels the best case (lowest total $\Phi$) is shown in red.

**Table 1.** CABLE parameters to be tuned.

| Parameter | Description |
| --- | --- |
| alloclg, alloclw | Allocation of C to leaves (grassy and woody) |
| ratiojv | Jmax/Vcmax |
| fsatmax | Multiplier for litter depth |
| dleaf_g, dleaf_w | Leaf length (grassy and woody) |
| vcmax_g, vcmax_w | Maximum RuBP carboxylation rate to leaf (grassy and woody) |
| hc_g, hc_w | Canopy height (grassy and woody) |
| f10_g, f10_w | Fraction of roots in top 10 cm (grassy and woody) |
| zr_g, zr_w | Maximum rooting depth (grassy and woody) |
| lgamma_g, lgamma_w | (log10 of) parameter in root efficiency function (grassy and woody) |
| a1 | Parameter in stomatal conductance function |
| ds0 | Sensitivity of stomatal conductance to VPD |

**Table 2.** CASA-CNP parameters to be tuned.

| Parameter | Description |
| --- | --- |
| soilc0_frac | Fraction of soil C in top 15 cm |
| age_leaf_g, age_leaf_w | Leaf turnover time (grassy and woody) |
| age_wood | Woody biomass turnover time (yr) |
| age_clitt1 | Base metabolic litter turnover time (yr) |
| age_clitt2 | Base fine structural litter turnover time (yr) |
| age_clitt3 | Base coarse woody debris turnover time (yr) |
| age_csoil1 | Fast soil C pool turnover time (yr) |
| age_csoil2 | Slow soil C pool turnover time (yr) |
| age_csoil3 | Passive soil C pool turnover time |
| fallocc_w | Fraction of non-leaf C allocated to wood |
| rsratio_g, rsratio_w | Fine root to shoot ratio (grassy and woody) |
| $q,c,w_0,w_1,w_2,w_3$ | Six parameters to define function for effect of soil moisture on soil respiration |

**Table 3.** Location and type of vegetation at the OzFlux sites (Beringer et al. (2016a); Isaac et al. (2016), http://www.ozflux.org.au) used in this study. Data period is for all of the data we have used at each site, with a subset of this data used for calibration.

| Site | Coordinates | Ecosystem | Data period | Calibration period | Reference |
|---|---|---|---|---|---|
| 1. Howard Springs | 12.4952$^o$S, 131.1501$^o$E | Open woodland savanna | 01/2001–12/2013 | 2001–2011 | Hutley et al. (2005) Beringer et al. (2011) |
| 2. Adelaide River | 13.0769$^o$S, 131.1178$^o$E | Savanna | 01/2007–05/2009 | - | Beringer et al. (2007) Beringer et al. (2011) |
| 3. Daly River Savanna | 14.1592$^o$S, 131.3833$^o$E | Woodland savanna | 01/2007–12/2013 | 2007–11 | Beringer et al. (2011) |
| 4. Dry River | 15.2588$^o$S, 132.3706$^o$E | Open forest savanna | 07/2008–12/2013 | 2008–11 | Beringer et al. (2011) |
| 5. Sturt Plains | 17.1507$^o$S, 133.3502$^o$E | Open grassland | 01/2008–12/2013 | - | Beringer et al. (2011) |
| 6. Alice Springs Mulga | 22.283$^o$S, 133.249$^o$E | Mulga woodland | 09/2010–12/2013 | 2010–2013 | Cleverly et al. (2013) |
| 7. Great Western Woodland | | Woodland | 01/2013–10/2013 | - | |
| 8. Gingin (Gnangara) | 31.3764$^o$S, 115.7139$^o$E | Banksia woodland | 01/2011–11/2013 | - | |
| 9. Calperum | 34.0027$^o$S, 140.5877$^o$E | Mallee | 01/2010–10/2013 | 2010–2013 | Meyer et al. (2015) |
| 10. Tumbarumba | 35.6566$^o$S, 148.1517$^o$E | Cool temperate wet sclerophyll | 01/2001–12/2013 | 2004–2010 | Leuning et al. (2005) van Gorsel et al. (2007) |
| 11. Nimmo Plains | 36.2159$^o$S, 148.5528$^o$E | Grassland | 01/2007–12/2013 | 2007–2011 | |
| 12. Whroo | 36.6731$^o$S, 145.0262$^o$E | Woodland | 12/2011–12/2013 | - | Beringer (2013a) |
| 13. Wombat | 37.4222$^o$S, 144.0944$^o$E | Cool temperate dry sclerophyll | 01/2010–12/2013 | 2010–2013 | |
| 14. Wallaby Creek | 37.4262$^o$S, 145.1872$^o$E | Old growth temperate | 08/2005–01/2009 | - | Martin et al. (2007) Beringer (2013b) |