# Peer review of "Interannual variability in Australia's terrestrial carbon cycle constrained by multiple observation types"

_Biogeosciences, 2016_

## Referee Comment (RC1) · Anonymous Referee #1 · 1 Jun 2016

Trudinger et al. reported a study on the interannual variability of terrestrial carbon fluxes in Australia using a model-data fusion approach. Comparing to previous studies, this work used a multiple constraints approach and explored the uncertanty from parameter equifinality. It is an interesting and useful study, however I have a few major concerns on the methods and manuscript organization:

Major points (1) The models (CABLE and CASA-CNP) produce daily carbon and water fluxes, why do the authors used monthly mean flux measurements (P5 L16) as constraints? Also, please make a table for the other biometric data (e.g., leaf NPP, soil carbon and above-ground phytomass and litter etc.) if there are a few data points. (2) Please provide more details on the calculation of the cost function. For example, how

"Different types of observations were then weighted relative to each other"? (3) The approach of generating ensemble parameter sets is not clear. How does "the null space Monte Carlo method" work? The authors stated that the purpose of using this method is "to quantify uncertainty due to parameter equifinality in model predictions", so my understanding is that this method can generate the posterior parameter distributions. Although the authors have detailed introduction to this method (P6 L27-33), I feel it is still difficult for the readers to understand why the generated parameter sets are sufficient to represent the posterior distribution of the parameters. Are there some special features of this method? Otherwise I can't believe only 30 parameter sets are enough to represent possible combination of parameters that are consistent with the observations, given more than 10 parameters are involved in each model. Even if we assume there are only 10 parameters, and each parameter can only be two possible values, there will 2^10 possible parameter sets. I understand that it may not be feasible to run regional simulations for a huge number of sets of parameters, but the authors need to demonstrate the 30 parameter sets are a good sample of the posterior parameter space.

Specific points: P1,L15, L16: ecosystem respiration –> heterotrophic respiration P1,L21: you can give a hint to the readers that the detailed description of BIOS-2 is in setion 2.1 P5, L12, L25: evaporation –> evapotranspiration P8, L7: R2 is unitless Fig 6, Fig S11: change the figure style to 2D. i.e., x axis is parameter, y axis is observation variable (ET, NPP etc), and use colors (light to dark) to represent the increase/decrease in variance P9, L12: please give the sources of the six bioclimatic regions P9, L18: the anomaly from the best case

consider citing these papers:

Richardson, A.D., M. Williams, D.Y. Hollinger, D.J.P. Moore, D.B. Dail, E.A. Davidson, N.A. Scott, R.S. Evans, H. Hughes, J.T. Lee, C. Rodrigues, and K. Savage. 2010. Estimating parameters of a forest ecosystem C model with measurements of stocks and fluxes as joint constraints. Oecologia, 164: 25-40, doi: 10.1007/s00442-010-1628-

y Chen, M., Zhuang, Q., Cook, D. R., Coulter, R., Pekour, M., Scott, R. L., Munger, J. W., and Bible, K.: Quantification of terrestrial ecosystem carbon dynamics in the conterminous United States combining a process-based biogeochemical model and MODIS and AmeriFlux data, Biogeosciences, 8, 2665-2688, doi:10.5194/bg-8-2665-2011, 2011. Keenan, T.F., E. Davidson, A. Moffat, W. Munger, and A.D. Richardson. 2012. Using model-data fusion to interpret past trends, and quantify uncertainties in future projections, of terrestrial ecosystem carbon cycling. Global Change Biology, 18: 2555-2569, doi: 10.1111/j.1365-2486.2012.02684.x

---

## Referee Comment (RC2) · Anonymous Referee #2 · 19 Jun 2016

Trudinger et al. present a modeling study investigating the range in inter-annual anomalies (or inter-annual variability – IAV) in net ecosystem production (NEP) in Australia due to parameter uncertainty and some further adaptations to the existing model set-up, namely a new function for the impact of soil moisture on soil respiration.

The work is extensive, interesting and of relevance to the community, especially given the recent finding that semi-arid regions are likely the dominant driver of the IAV and trend in the global C sink, as the authors detail nicely in the introduction. One of the main conclusions that the authors highlight, that there may be a range of values for the NEP anomalies that are dependent on parameter uncertainty, and therefore that we cannot just use one optimised parameter set, is an important point to make.

However, I have several concerns about the description of the overall objectives, and how the experiments link to these objectives, that I have laid out below. In my comments I have tried made some suggestions that I feel would improve the manuscript in these respects.

The objectives could be more clearly laid out in the introduction, especially given the number of different tests (forcing, model structure and optimisation set-up with different observations) that are then detailed, which I feel could lead to the reader feeling confused as to what the overall aims are. For example, stating (in the introduction), "...we use BIOS-2 to explore inter-annual variability in NEP..." lacks some detail. "We are also interested in the effect of uncertainty in model parameters on modelled IAV" is more clear, but it might be better to formulate some key objectives as bullet points or questions and then directly link the results to these objectives.

I like the description of the issues related to parameters that can and cannot be well constrained by the optimisation in Section 2.4 (p6 lines ∼10-20). And I think the aim of producing a range of estimates of NEP anomalies based on parameter uncertainty/equifinality is a good one. However I think it is important to detail what you mean by "parameters that are not informed by the calibration dataset". Equifinality can occur for two reasons (broadly speaking): i) parameters to which the model variables used in the optimisation are not sensitive (therefore will not be constrained by the optimisation), and ii) parameters to which the model variables are sensitive, but which are correlated with other parameters. In this latter case it is possible that the parameters will be well constrained, but to the "wrong" values (i.e. if you did a test with psuedo observations where you knew the right value for the parameters, the posterior value would be well-constrained but not correct, possibly due to the particular noise realisations in the given set of observations. In my understanding, parameter equifinality mostly Are you trying to account for both types in your ensembles? From p6 lines 28-29 (point i) it seems that perhaps just the first type? Also, it isn't clear to me that you are also accounting for uncertainty that still remains in parameters that have been well-constrained (but not

perfectly) by the optimisation – i.e. calculated from the posterior parameter covariance matrix. Given these questions, I think the manuscript would benefit considerably from a more detailed description of exactly what uncertainty is represented in the parameter ensembles that are generated using the null point Monte Carlo method (p6 lines ∼21-34).

Given your focus on the inter-annual anomalies, I think it would be good to have a plot showing the inter-annual variability of the observations, prior and posterior, perhaps as a bar graph of mean year to year anomalies, in order to asses how (if) the optimisation improves the modeling of the inter-annual anomalies. It is not very easy to evaluate this from the annual time series plots, and it is not shown on Figure 9. It is not clear that the simulations of the IAV are improved by all this assimilation work, which was a question I had when I started reading the paper, given previous studies have shown that parameter optimisation can improve the fit to seasonal fluxes but not necessarily to the IAV (Kuppel et al. 2012). Indeed on page 11 Line24 you state that the NEP anomalies (the key topic under investigation in this study) are not significantly improved by the optimisation, yet the majority of the paper is about the different optimisation tests, rather than an investigation of the IAV simulated by the model after the optimisation. Therefore at times I felt like I was reading a study about different optimisation configurations, and the link to the IAV was missing (until the discussion). There is a lot of information in Figures 9, S14 and 15 that has not been discussed and could contribute greatly to the discussion on IAV. The authors mention that it appears that NPP is the main driver but most of the uncertainty is in the heterotrophic respiration. This is a useful conclusion but I feel a greater discussion of this result would add value to this study (I have made further comments on this in the minor comments below).

Given that the optimisation does not dramatically improve the IAV I was left feeling the paper was somewhat disjointed in its objectives. Also, the fact that the optimisation does not improve the simulation of anomalies is a key finding and could be discussed more. What could the modeling community do to better optimise the model/simulate

the NEP anomalies? Some of the tests, for example the focus on the soil moisture affect on soil respiration, the contribution of different data streams, are not well linked to the initial objectives that are laid out. For example it was unclear to me why there is a focus on the soil respiration function despite the study of Exbrayat et al. (2013), particularly because in the introduction the authors discuss that previous studies have shown that NPP is the key driver. While obviously it would appear this function is important to investigate in terms of the development of the model in general, it was not clear that this was crucial for this study investigating the NEP anomalies. I would detail more clearly up front (in the introduction) why you have included different aspects (including the soil respiration function) that haven't been included in previous modeling studies with BIOS-2, i.e. explain (as you do later in the discussion p11 lines 30-35), that the purpose is to have best model set-up, otherwise I'm searching for a reason as to why these tests are particularly important for simulating the NEP anomalies.

This is maybe a style issue, but to deal with the issues raised in the previous comments, I think it may be better to merge some of the discussion into the results. I found the "story" of the manuscript came together more in the discussion, but the description of results seemed incomplete at times or in need of interpretation. As already mentioned I was left wondering how each section (that may have contained a very nice test in itself) fitted in with the overall goals of better quantifying the NEP anomalies, instead of just being an interesting parameter optimisation test bed.

Minor comments

Why did you use monthly observations when I presume daily (or half hourly) fluxes are available? I guess this is because you are focusing on seasonal to annual (inter-annual) timescales but this needs a justification. Out of interest have you done a test with daily observations to see if the optimisation of IAV is improved? Similarly, why were long-term means used for streamflow data?

How were the observation error correlations taken into account? This will be a particular issue for GPP observations that were derived from the NEP eddy covariance data via a flux partitioning method.

Figure 2: it would be informative to see prior and posterior RMSE or R, some metric to show the improvement.

Does Figure 4 only show the posterior results?

It might be good to put which observations are used to optimise the CABLE and CASA-CNP parameters in Table 1 and 2.

Section 2.3: do you mean evaporation or evapotranspiration measurements were used? In the figures you say "ET", which is often used to denote the latter.

P5 Line 30: define the PEST acronym (it might also be better to put the website here instead of later).

Section 2.4: it might be good to describe briefly what the "down-gradient" method is for the non-specialist.

P6 Line 21: How did you define the larger number of parameters from the initial SA used in the Haverd et al. (2013b) study?

P6 Line 35 to next page: apologies if I've missed this, but why 21 parameter sets for CABLE and 19 for CASA-CNP?

P7 Line 35: What does this tell us? It shouldn't be the case that an optimisation with one data stream degrades the fit to another if the models are consistent with each other and with the observations, and the prior error covariance matrix are properly characterised.

P8 Line 5: "likely attributable" → can you do a test to quantify this further, given the focus on this new function?

Figure 6b: Is this for each observation type when it is used on its own, or when it is left

out (as I understand for Figure 6a)?

As mentioned above, Section 3.2 is one example of a section that lacks some more detailed description, rather than just a summary and explanation of the figures and how to interpret them (which is also useful). Aside from NPP, which observations are useful for which processes (even qualititatively, or using a summary metric), particularly in reference to constraining the IAV? There is also no description of Figure S11.

Figure 7: is this the total model-data mismatch using all observations?

P9 Line 1: parameter identifiaility → I like this analysis, but one could also simply look at the parameter posterior covariance matrix which may be easier for some readers to follow as an initial example of parameter correlation.

P9 Line 10: Figures 12 and 13, not 11 and 12.

P10 Line 1: We don't know that there is only a little difference in the misfit function (when looking at fig. 11a) if we don't know what the prior misfit is.

P10 Lines 13-15 (Section 3.5): would be good to have a reference to a figure or table here.

P11 Line 16: Is it really clear or fair to say that the parameters related to heterotrophic respiration are particularly important when you have introduced a new function for the impact of soil moisture on respiration which has quite a number of parameters? Is it surprising that you have more equifinality for heterotrophic respiration given this new function? It would be good to describe this more in the results because at this point I was wondering if this was a clear conclusion of the analysis (i.e. that the heterotrophic respiration-related parameters are particularly important). A link to a figure at this point in the discussion may also help.

P11 Lines 21-22: Similar point to above. It is not immediately obvious to me, looking at Figure 9, S14 and S15 that the NEP anomaly is dominated by equifinality in the heterotrophic respiration. It depends which ecosystem contributes to the anomaly,

which was only briefly discussed. Looking at Australia overall for 2011, it looks like the uncertainty due to parameter equifinality is the roughly the same for NPP and heterotrophic respiration. I think the same can be said for the ecosystems the authors say below contribute most to the NEP anomalies (savanna and sparse vegetation).

In general all the routines used from PEST should be described in more detail somewhere (supplementary?).

Kuppel, S., Peylin, P., Chevallier, F., Bacour, C., Maignan, F., and Richardson, A. D.: Constraining a global ecosystem model with multi-site eddy-covariance data, Biogeosciences, 9, 3757-3776, doi: 10.5194/bg-9-3757-2012.
* * *

---

## Author Comment (AC1) · 1 Aug 2016

We thank reviewer 1 for their helpful comments, and provide the following responses:

**(1) The models (CABLE and CASA-CNP) produce daily carbon and water fluxes, why do the authors used monthly mean flux measurements (P5 L16) as constraints? Also, please make a table for the other biometric data (e.g., leaf NPP, soil carbon and above-ground phytomass and litter etc.) if there are a few data points.**

We use monthly mean flux measurements as constraints because the meteorological forcing is not as accurate at the daily timescale. There are too many data points for a

table, instead we include map (Fig. 1 below) showing the location of the biometric and other data that are used for calibration.

**(2) Please provide more details on the calculation of the cost function. For example, how "Different types of observations were then weighted relative to each other"?**

The weights for each observation group (e.g. ET, GPP etc) were scaled so that each group contributed approximately equally to the cost function calculated with the prior parameters. This is important because the different types of observations can have vastly different magnitudes, and the relative contribution of each group to the cost function should not depend on the units that are used.

**(3) The approach of generating ensemble parameter sets is not clear. How does "the null space Monte Carlo method" work? The authors stated that the purpose of using this method is "to quantify uncertainty due to parameter equifinality in model predictions", so my understanding is that this method can generate the posterior parameter distributions. Although the authors have detailed introduction to this method (P6 L27-33), I feel it is still difficult for the readers to understand why the generated parameter sets are suffi- cient to represent the posterior distribution of the parameters. Are there some special features of this method? Otherwise I can't believe only 30 parameter sets are enough to represent possible combination of parameters that are consistent with the observations, given more than 10 parameters are involved in each model. Even if we assume there are only 10 parameters, and each parameter can only be two possible values, there will $2^{10}$ possible parameter sets. I understand that it may not be feasible to run regional simulations for a huge number of sets of parameters, but the authors need to demonstrate the 30 parameter sets are a good sample of the posterior parameter space.**

The null space Monte Carlo method does not calculate posterior parameter distributions. Rather, it is an efficient method to generate multiple parameter sets that are constrained by the calibration observations (Tonkin and Doherty, 2009). By taking into account the calibration and null spaces, it allows for these parameter sets to have the most range in combinations of parameters that are least well constrained by the calibration observations, therefore allowing for a much better representation of parameter equifinality than random combinations of parameters. Clearly, the more parameter sets the better, but we need to balance that with the computational limitations. Figures 7 and S12 show the ensembles of parameter sets (colored symbols), and they have many different values for the parameters that are not well constrained (many more than two values). Our purpose of using this method would perhaps be better described as "to see the effect of uncertainty due to parameter equifinality in model predictions".

**Specific points: P1,L15, L16: ecosystem respiration –> heterotrophic respiration**

Yes, thank you.

**P1,L21: you can give a hint to the readers that the detailed description of BIOS-2 is in section 2.1**

Yes

**P5, L12, L25: evaporation –> evapotranspiration**

Yes, thank you.

**P8, L7: R2 is unitless**

Yes

**Fig 6, Fig S11: change the figure style to 2D. i.e., x axis is parameter, y axis is observation variable (ET, NPP etc), and use colors (light to dark) to represent the in- crease/decrease in variance**

Presumably the reason for changing the figure is that some bars were obscured from view. We have changed the figure to 2D, but still use bars rather than colors, and all

bars can be clearly seen now. New figures for observation worth for CABLE (Fig. 2 below) and CASA-CNP (Fig. 3 below) replace the previous Figures 6 and S11.

**P9, L12: please give the sources of the six bioclimatic regions**

The bioclimatic regions are an aggregation of the agro-climatic classification of Hutchinson et al. (2005) into six classes, as described and used by Haverd et al (2013a, 2013b).

**P9, L18: the anomaly from the best case**

No, the anomaly is calculated for each ensemble member by subtracting the temporal average for that ensemble member.

References:

Hutchinson, M. F., McIntyre, S., Hobbs, R. J., Stein, J. L., Garnett, S., and Kinloch, J.: Integrating a global agro-climatic classification with bioregional boundaries in Australia, Global Ecology and Biogeography, 14, 197–212, doi:10.1111/j.1466-822X.2005.00154.x, 2005.

Tonkin, M., and J. Doherty, Calibration-constrained Monte Carlo analysis of highly parameterized models using subspace techniques, Water Resour. Res., 45, W00B10, doi:10.1029/2007WR006678, 2009.

[Figure]

[Figure]

**OzFlux sites**
**Above-ground phytomass**
**Above-ground litter**
**Leaf NPP**
**Soil carbon**
**Steamflow in unimpaired catchments**

**Fig. 1.**

**Fig. 2.**

[Figure]

Fig. 3.

---

## Author Response (AR1)

**Author response to reviewer comments and manuscript changes in "Interannual variability in Australia's terrestrial carbon cycle constrained by multiple observation types", by Trudinger et al**

Reviewers Comments [RC], Authors' Response [AR] and Manuscript Changes [MC] are are follows:

**Author response to comments by Reviewer #1**

We thank reviewer 1 for their helpful comments, and provide the following responses:

**[RC] (1) The models (CABLE and CASA-CNP) produce daily carbon and water fluxes, why do the authors used monthly mean flux measurements (P5 L16) as constraints? Also, please make a table for the other biometric data (e.g., leaf NPP, soil carbon and above-ground phytomass and litter etc.) if there are a few data points.**

[AR] We use monthly mean flux measurements as constraints because the meteorological forcing is not as accurate at the daily timescale. There are too many data points for a table, instead we include map showing the location of the biometric and other data that are used for calibration.

*[MC] Add the following to Section 2.3 on Observations "Although the model produces daily carbon and water fluxes, we used monthly rather than daily flux observations because the meteorological forcing is not as reliable at the daily timescale." Add a new figure (new Fig 2) that is a map of observation locations.*

**[RC] (2) Please provide more details on the calculation of the cost function. For example, how "Different types of observations were then weighted relative to each other"?**

[AR] The weights for each observation group (e.g. ET, GPP etc) were scaled so that each group contributed approximately equally to the cost function calculated with the prior parameters. This is important because the different types of observations can have vastly different magnitudes, and the relative contribution of each group to the cost function should not depend on the units that are used.

*[MC] Add the text "The weights ..." from [AR] to Section 2.4 Optimisation method.*

**[RC] (3) The approach of generating ensemble parameter sets is not clear. How does "the null space Monte Carlo method" work? The authors stated that the purpose of using this method is "to quantify uncertainty due to parameter equifinality in model predictions", so my understanding is that this method can generate the posterior parameter distributions. Although the authors have detailed introduction to this method (P6 L27-33), I feel it is still difficult for the readers to understand why the generated parameter sets are suffi- cient to represent the posterior distribution of the parameters. Are there some special features of this method? Otherwise I can't believe only 30 parameter sets are enough to represent possible combination of parameters that are consistent with the observations, given more than 10 parameters are involved in each model. Even if we assume there are only 10 parameters, and each parameter can only be two possible values, there will 210 possible parameter sets. I understand that it may not be feasible to run regional simulations for a huge number of sets of parameters, but the authors need to demonstrate the 30 parameter sets are a good sample of the posterior parameter space.**

[AR] The null space Monte Carlo method does not calculate posterior parameter distribu- tions. Rather, it is an efficient method to generate multiple parameter sets that are constrained by the calibration observations (Tonkin and Doherty, 2009). By taking into account the calibration and null spaces, it allows for these parameter sets to have the most range in combinations of parameters that are least well constrained by the calibration observations, therefore allowing for a much better representation of parameter equifinality than random combinations of parameters. Clearly, the more parameter sets the better, but we need to balance that with the computational limitations. Figures 7 and S12 show the ensembles of parameter sets (colored symbols), and they have many different values for the parameters that are not well constrained (many more than two values). Our purpose of using this method would perhaps be better described as "to see the effect of uncertainty due to parameter equifinality in model predictions".

*[MC] We change "quantify uncertainty due to parameter equifinality" to "see the effect of uncertainty due to parameter equifinality". We add two additional references, Tonkin and Doherty (2009) and Sepulveda and Doherty (2015) that give further details about the method. We also add "The NSMC method does not specifically calculate the posterior parameter probability distributions, however it is an efficient way to generate multiple parameter sets that span a significant amount of the equifinality in both models."*

Specific points:

**[RC] P1,L15, L16: ecosystem respiration $->$ heterotrophic respiration**

[AR] Yes, thank you.

*[MC] Change "ecosystem respiration" to "heterotrophic respiration" at the two locations on page 1*

**[RC] P1,L21: you can give a hint to the readers that the detailed description of BIOS-2 is in section 2.1**

[AR] Yes

*[MC] Add "described in detail in Section 2.1" to the Introduction.*

**[RC] P5, L12, L25: evaporation - evapotranspiration**

[AR] Yes, thank you.

*[MC] Change "evaporation" to "evapotranspiration" or "ET" in two places in Section 2.3 and Figure captions.*

**[RC] P8, L7: R2 is unitless**

[AR] Yes

*[MC] Remove units.*

**[RC] Fig 6, Fig S11: change the figure style to 2D. i.e., x axis is parameter, y axis is observation variable (ET, NPP etc), and use colors (light to dark) to represent the in- crease/decrease in variance**

[AR] Presumably the reason for changing the figure is that some bars were obscured from view. We have changed the figure to 2D, but still use bars rather than colors, and all bars can be clearly seen now. New figures for observation worth for CABLE and CASA-CNP replace the previous Figures 6 and S11.

*[MC] Change figures 6 (now 7) and S11 as described in [AR].*

**[RC] P9, L12: please give the sources of the six bioclimatic regions**

[AR] The bioclimatic regions are an aggregation of the agro-climatic classification of Hutchinson et al. (2005) into six classes, as described and used by Haverd et al (2013a, 2013b).

*[MC] Text in [AR] added to Section 3.2.*

**[RC] P9, L18: the anomaly from the best case**

[AR] No, the anomaly is calculated for each ensemble member by subtracting the temporal average for that ensemble member.

*[MC] Add "When anomalies are shown for ensemble members here and in subsequent figures, they are calculated for each ensemble member by subtracting the temporal average of the quantity for that ensemble member." to Section 3.1.1.*

**[RC] Consider citing these papers: Richardson et al., (2010), Chen et al., (2011), Keenan et al., (2012).**

[AR] Thanks to the reviewer for drawing these to our attention.

*[MC] We have added citations for all of these papers.*

**Author response to comments by Reviewer #2**

We thank reviewer #2 for their helpful comments, and provide the following responses:

**[RC] The objectives could be more clearly laid out in the introduction, especially given the number of different tests (forcing, model structure and optimisation set-up with different observations) that are then detailed, which I feel could lead to the reader feeling confused as to what the overall aims are. For example, stating (in the introduction), "...we use BIOS-2 to explore inter-annual variability in NEP..." lacks some detail. "We are also interested in the effect of uncertainty in model parameters on modelled IAV" is more clear, but it might be better to formulate some key objectives as bullet points or questions and then directly link the results to these objectives.**

[AR] We agree that the objectives should be more clearly stated, and suggest the following: The objectives of this study are to use multiple observation types to constrain the IAV of terrestrial carbon fluxes for Australia. Specifically, multiple observation types are used to optimise parameters in BIOS-2.1, by generating an ensemble of acceptable parameter sets that will allow us to see the effect of parameter equifinality. We then use these parameter sets in the model to calculate IAV in Australian NEP over recent decades. We are interested in the following questions: What is our best estimate of IAV in Australian carbon fluxes? How does parameter equifinality affect modelled estimates of IAV and the 2011 anomaly for Australia? How does parameter equifinality effect estimates of the processes contributing to IAV in NEP, including NPP and heterotrophic respiration and the effect of soil moisture on heterotrophic respiration?

*[MC] Text added to Introduction as described in [AR].*

**[RC] I like the description of the issues related to parameters that can and cannot be well constrained by the optimisation in Section 2.4 (p6 lines ~10–20). And I think the aim of producing a range of estimates of NEP anomalies based on parameter**

uncertainty/equifinality is a good one. However I think it is important to detail what you mean by "parameters that are not informed by the calibration dataset". Equifinality can occur for two reasons (broadly speaking): i) parameters to which the model variables used in the optimisation are not sensitive (therefore will not be constrained by the optimisation), and ii) parameters to which the model variables are sensitive, but which are correlated with other parameters. In this latter case it is possible that the parameters will be well constrained, but to the "wrong" values (i.e. if you did a test with psuedo observations where you knew the right value for the parameters, the posterior value would be wellconstrained but not correct, possibly due to the particular noise realisations in the given set of observations. In my understanding, parameter equifinality mostly Are you trying to account for both types in your ensembles? From p6 lines 28-29 (point i) it seems that perhaps just the first type? Also, it isn't clear to me that you are also accounting for uncertainty that still remains in parameters that have been well-constrained (but not perfectly) by the optimisation " i.e. calculated from the posterior parameter covariance matrix. Given these questions, I think the manuscript would benefit considerably from a more detailed description of exactly what uncertainty is represented in the parameter ensembles that are generated using the null point Monte Carlo method (p6 lines 21-34).

[AR] We will add the following: The ensemble from the Null Space Monte Carlo analysis will include the effect of uncertainty in parameters to which model outputs for comparison with observations are not sensitive, as well as parameters to which the model outputs are sensitive but which are correlated with other parameters, as both of these are part of the calibration null space. In addition, the recalibration process and the fact that solutions with a range of values of Phi are retained means that the ensemble also accounts for uncertainty in parameters that are well constrained by observations but affected by measurement noise or model structural error (Sepúlveda and Doherty, 2015).

[MC] Text added to Section 2.4 as described in [AR].

[RC] Given your focus on the inter-annual anomalies, I think it would be good to have a plot showing the inter-annual variability of the observations, prior and posterior, perhaps as a bar graph of mean year to year anomalies, in order to asses how (if) the optimisation improves the modeling of the inter-annual anomalies. It is not very easy to evaluate this from the annual time series plots, and it is not shown on Figure 9. It is not clear that the simulations of the IAV are improved by all this assimilation work, which was a question I had when I started reading the paper, given previous studies have shown that parameter optimisation can improve the fit to seasonal fluxes but not necessarily to the IAV (Kuppel et al. 2012). Indeed on page 11 Line24 you state that the NEP anomalies (the key topic under investigation in this study) are not significantly improved by the optimisation, yet the majority of the paper is about the different optimisation tests, rather than an investigation of the IAV simulated by the model after the optimisation. Therefore at times I felt like I was reading a study about different optimisation configurations, and the link to the IAV was missing (until the discussion). There is a lot of information in Figures 9, S14 and 15 that has not been discussed and could contribute greatly to the discussion on IAV. The authors mention that it appears that NPP is the main driver but most of the uncertainty

is in the heterotrophic respiration. This is a useful conclusion but I feel a greater discussion of this result would add value to this study (I have made further comments on this in the minor comments below).

[AR]

- Suggested figure and assessing whether optimisation improves IAV - The question of whether optimisation improves IAV isn't so important, as it depends on what parameters you start with. Some of the parameters were already optimised in Haverd et al. (2013a). We were more interested in the uncertainty due to parameter equifinality, as well as how well we could model IAV at the flux sites (the answer to this question is, not particularly well). There are already timeseries plots, scatter plots and statistics showing how well we model IAV at the flux sites, so we don't believe an additional plot as suggested is warranted. In addition, the flux records at some of the sites only have a few years of data, so we wouldn't want to calculate mean year to year anomalies for all sites.

- By merging some of the Discussion into the Results section, as mentioned below, we hope to keep the focus on IAV and consolidate most of the discussion of Figs 9, S14 and S15.

- The conclusion of greater uncertainty in heterotrophic respiration - see comments below.

*[MC] We have not added the suggested figure. The other points are addressed below.*

[RC] Given that the optimisation does not dramatically improve the IAV I was left feeling the paper was somewhat disjointed in its objectives. Also, the fact that the optimisation does not improve the simulation of anomalies is a key finding and could be discussed more. What could the modeling community do to better optimise the model/simulate the NEP anomalies? Some of the tests, for example the focus on the soil moisture affect on soil respiration, the contribution of different data streams, are not well linked to the initial objectives that are laid out. For example it was unclear to me why there is a focus on the soil respiration function despite the study of Exbrayat et al. (2013), particularly because in the introduction the authors discuss that previous studies have shown that NPP is the key driver. While obviously it would appear this function is important to investigate in terms of the development of the model in general, it was not clear that this was crucial for this study investigating the NEP anomalies. I would detail more clearly up front (in the introduction) why you have included different aspects (including the soil respiration function) that haven't been included in previous modelling studies with BIOS-2, i.e. explain (as you do later in the discussion p11 lines 30-35), that the purpose is to have best model set-up, otherwise I'm searching for a reason as to why these tests are particularly important for simulating the NEP anomalies.

[AR]

- The objectives need to be more clearly stated, see above.

- We will put more emphasis on the point that although the model does not match IAV well at the flux sites, measurements at most of the Ozflux sites do not show a strong relationship between NEP and available soil water, and therefore are not particularly representative of IAV in NEP for Australia as a whole.

- What could the modelling and observation community do to improve simulation of IAV - We

suggest the following: Flux observations at more representative sites might help. It is also not clear whether the meteorological drivers can explain the IAV at the current flux sites, and a study similar to Abramowitz et al (2008) using statistical models but focussed on the interannual timescale at Australian sites may be useful to answer that question.

- Why focus on soil respiration function - yes, we did want to have the best model set-up. Soil moisture is important for soil respiration, and precipitation is important for IAV in NEP, so we wanted to ensure that we had the best estimate of the timing of heterotrophic respiration, and to have uncertainty in this function contribute to parameter uncertainty in NEP IAV. We're not aware of previous studies that have formally optimised this function. These points would be included in a revised manuscript.

*[MC] Clearly state the objectives, as described above. Add to the Conclusions "The timing of interannual variations in NEP at the flux sites is not particularly well captured by the model, as has been found in previous modelling studies, however most of the flux measurements are from locations that are not water limited, in contrast to the parts of the country that most influence Australian NEP.". Add text "Flux observations at more ..." from [AR] to the Discussion. Add to the Introduction "We include some improvements to the model structure and forcing data (specifically, ..."*

**[RC] This is maybe a style issue, but to deal with the issues raised in the previous comments, I think it may be better to merge some of the discussion into the results. I found the 'story' of the manuscript came together more in the discussion, but the description of results seemed incomplete at times or in need of interpretation. As already mentioned I was left wondering how each section (that may have contained a very nice test in itself) fitted in with the overall goals of better quantifying the NEP anomalies, instead of just being an interesting parameter optimisation test bed.**

[AR] We agree that it would help the flow of the manuscript to merge some of the discussion into the results.

*[MC] We have merged a significant amount of the information from the Discussion into the Results.*

**Minor comments**

**[RC] Why did you use monthly observations when I presume daily (or half hourly) fluxes are available? I guess this is because you are focusing on seasonal to annual (interannual) timescales but this needs a justification. Out of interest have you done a test with daily observations to see if the optimisation of IAV is improved? Similarly, why were long-term means used for streamflow data?**

[AR] We use monthly mean flux measurements as constraints because the meteorological forcing is not as accurate at the daily timescale, and not at all accurate at subdaily timescales (we use daily meteorological forcing downscaled to the CABLE timestep using a weather generator, as mentioned in Section 2.1). We have not tested the optimisation with daily observations, and would not expect an improvement. Long-term means are used for streamflow observations because BIOS-2 does not model streamflow dynamics well, something that we plan to address in future work. This information will be added.

*[MC] Add the following to Section 2.3 on Observations "Although the model produces daily carbon and water fluxes, we used monthly rather than daily flux observations because the meteorological*

*forcing is not as reliable at the daily timescale."* and *"Long-term means are used for streamflow observations because BIOS-2 does not model streamflow dynamics well, something that we plan to address in future work."*

**[RC] How were the observation error correlations taken into account? This will be a particular issue for GPP observations that were derived from the NEP eddy covariance data via a flux partitioning method.**

[AR] We did not take into account observation error correlations. We acknowledge that GPP and NEP would have correlated errors, but we don't have good information about their error statistics. Temporal correlations have been shown (Lasslop et al., 2008) to decrease significantly with increasing time lags, so we don't expect temporal correlations to be particularly important for monthly measurements.

*[MC] No change made.*

**[RC] Figure 2: it would be informative to see prior and posterior RMSE or R, some metric to show the improvement.**

[AR] We add the following: Phi for the best case divided by Phi for prior parameters, split into observation groups, is as follows: ET 0.88, GPP 0.46, NPP 0.08 and streamflow 1.06 for CABLE observations and NEP 0.32, soil carbon 0.80, phytomass 0.95 and litter 0.78 for CASA-CNP observations. The best to prior ratio for total $\Phi$ was 0.36.

*[MC] Add text from [AR] to Section 3.1.1 Comparison of model outputs with observations.*

**[RC] Does Figure 4 only show the posterior results?**

[AR] Yes, this will be clarified.

*[MC] Add "(best case)" to figure caption (now Fig 5).*

**[RC] It might be good to put which observations are used to optimise the CABLE and CASACNP parameters in Table 1 and 2.**

[AR] This information is already given in the text at the end of Section 2.3, after the observations have been described. If the suggestion here is to add the types of observation to Tables 1 and 2, the reference to these tables comes in Section 2.2 before the observations are introduced, so we believe it is not beneficial to mention the observations in Tables 1 and 2.

*[MC] No change.*

**[RC] Section 2.3: do you mean evaporation or evapotranspiration measurements were used? In the figures you say "ET", which is often used to denote the latter.**

[AR] This should be evapotranspiration (ET) throughout.

*[MC] "Evaporation" changed to "evapotranspiration" or "ET"*

**[RC] P5 Line 30: define the PEST acronym (it might also be better to put the website here instead of later).**

[AR] Parameter ESTimation. Yes, the website should be mentioned here instead of later.

[MC] *"... we used the PEST implementation (Parameter ESTimation, http://www.pesthomepage.org)"*

**[RC] Section 2.4: it might be good to describe briefly what the "down-gradient" method is for the non-specialist.**

[AR] We would add the following: A down-gradient search method uses information about the gradient of the cost function with respect to the parameters to decide how to iteratively alter parameters to locate parameter values corresponding with the minimum in the cost function (Raupach et al., 2005).

[MC] *Add the text from [AR] to Section 2.4.*

**[RC] P6 Line 21: How did you define the larger number of parameters from the initial SA used in the Haverd et al. (2013b) study?**

[AR] Haverd et al excluded some parameters that were uncertain yet were identified in the sensitivity analysis as being unlikely to be constrained by the calibration observations. A number of these parameters were included here.

[MC] *Add to Section 2.4 "Haverd et al (2013b) used parameter sensitivity analysis to choose which parameters to optimise, avoiding parameters that were unlikely to be constrained by the available observations. Here we optimise a larger number of parameters, including some of the parameters that are not well constrained by the calibration dataset, to explore the effect of ..."*

**[RC] P6 Line 35 to next page: apologies if I've missed this, but why 21 parameter sets for CABLE and 19 for CASA-CNP?**

[AR] The case generated by the original optimisation of CABLE and CASA-CNP could be included in either ensemble. The numbers 21 and 19 counted this case in the CABLE ensemble, when it is actually more appropriate (and aligned with our original thinking) to count it in the CASA-CNP ensemble, giving 20 members each. The CASA-CNP ensemble is therefore all cases with the same CABLE parameters. This also reflects the colors that were used in figures such as Figure S12.

[MC] *Change text at the end of Section 2.4 to 20 parameter sets for each model.*

**[RC] P7 Line 35: What does this tell us? It shouldn't be the case that an optimisation with one data stream degrades the fit to another if the models are consistent with each other and with the observations, and the prior error covariance matrix are properly characterised.**

[AR] Although the fit to most observations was improved, the fit to a few observations was degraded. We are using a range of different types of observations of carbon and water in a complex model, so it is not entirely surprising that there are some discrepancies. Richardson et al (2010) pointed out that this often occurs. Nonetheless, this must be an indication of deficiencies in the model and/or the observations and their uncertainty characterisation, but we have not yet been able to identify the specific causes of these deficiencies in our model.

[MC] *Add text to Section 3.1.1 "The degradation of the fit to some observations is a consequence*

*of trying to fit many different types of observations at once with a complex model, and it is not entirely surprising that there are some discrepancies. Richardson et al (2010) pointed out that this often occurs. Nonetheless, it is an indication of deficiencies in the model, including the forcing and specification of parameters, and/or the observations and their uncertainty characterisation, but we have not yet been able to identify the specific causes of these deficiencies in our model."*

**[RC] P8 Line 5: "likely attributable" ! can you do a test to quantify this further, given the focus on this new function?**

[AR] This is difficult to test, given all of the other changes since the Haverd et al study. We therefore replace the text with "possibly attributable".

*[MC] Replace "likely attributable" with "possibly attributable".*

**[RC] Figure 6b: Is this for each observation type when it is used on its own, or when it is left out (as I understand for Figure 6a)?**

[AR] This is for each observation group when used on its own. This will be clarified.

*[MC] Add to each part of the figure caption, now figure 7, "(i.e. leaving out each group in turn)" and "(i.e. each observation group is used on its own)."*

**[RC] As mentioned above, Section 3.2 is one example of a section that lacks some more detailed description, rather than just a summary and explanation of the figures and how to interpret them (which is also useful). Aside from NPP, which observations are useful for which processes (even qualititatively, or using a summary metric), particularly in reference to constraining the IAV? There is also no description of Figure S11.**

[AR] We will include the following: Many of the CABLE parameters are constrained by more than one observation group. The eddy flux data (ET and GPP) provide the tightest constraints on the biophysical parameters, as also found by Haverd et al (2013a), presumably because they contain temporal information. Streamflow seems to contain mostly redundant information that is available from the other observations, but is still worth including to mitigate against the effect of biases in any single observation type. In future work, we plan to improve streamflow dynamics in the model, and would then hope to take advantage of temporal information in the streamflow measurements.

Figure S11 shows that many parameters are not well constrained by the calibration observations, and most of those that are constrained to some extent are influenced by only one observation group (e.g. age_leaf_w and age_clitt2 by litter, age_wood and falloc_w by phytomass and soilc0_frac, age_csoil1, age_csoil2 and age_csoil3 by soil carbon), demonstrating the benefit of including all of these observation types. The function describing the effect of soil moisture on soil respiration is constrained by observations of both NEP and soil carbon. This analysis of which observation groups constrain which parameters gives results that are mostly as we would have expected. However, we would have expected the soil respiration function to be constrained by litter observations, but this appears not to be the case, perhaps because the litter observations are quite sparse.

*[MC] Add text in [AR] to Section 3.1.1.*

**[RC] Figure 7: is this the total model-data mismatch using all observations?**

[AR] No, just the observations for CABLE. This will be clarified.

*[MC] Change figure caption (now Fig 8) to "Model-data mismatch for CABLE observations, $\Phi_{CABLE}, \ldots$"*

**[RC] P9 Line 1: parameter identifiaility ! I like this analysis, but one could also simply look at the parameter posterior covariance matrix which may be easier for some readers to follow as an initial example of parameter correlation.**

[AR] Parameter identifiability is based on analysis of the posterior parameter covariance matrix, including eigenvector analysis. While a simple look at the covariance matrix will tell you about parameter correlations, identifiability involves more sophisticated analysis too.

*[MC] Add to Section 3.1.1 "… parameter identifiability (Doherty and Hunt, 2009), based on analysis of the posterior parameter covariance matrix using tools that are available with PEST (routine IDENTPAR)*

**[RC] P9 Line 10: Figures 12 and 13, not 11 and 12.**

[AR] Yes, thank you.

*[MC] Figure numbers S11 and S12 changed to S12 and S13 in Section 3.1.1.*

**[RC] P10 Line 1: We don"t know that there is only a little difference in the misfit function (when looking at fig. 11a) if we don't know what the prior misfit is.**

[AR] The total cost function calculated with the prior parameters was 5411. However, the point we were trying to make here was that there seems to be no relationship between NEP IAV and Phi, as we see quite different values of 2011 NEP for very similar values of Phi.

*[MC] Add to Section 3.1.1 "For reference, $\Phi_{CABLE}$ with prior parameters was 3962" and "$\Phi_{CASA}$ for prior parameters (but using inputs from CABLE calculated with optimised parameters) was 1449." Rather than talking about "little difference in $\Phi$", we say "as we see quite different values of 2011 NEP for very similar values of total $\Phi$."*

**[RC] P10 Lines 13-15 (Section 3.5): would be good to have a reference to a figure or table here.**

[AR] This information is not shown clearly in any of the figures or tables, so we have added some numbers to the text as follows: Without the shade correction (Eq. 5), the agreement with calibration observations is a bit worse than our best case for some observation types (e.g. the ratio of optimised to prior Phi for the noshade case for NPP and soil carbon are 0.27 and 0.91, compared to 0.08 and 0.80 for our best case) and a bit better for others (GPP and NEP Phi ratio for the noshade case are 0.32 and 0.29, compared to 0.46 and 0.32 for our best case), but overall the total Phi is not significantly different.

*[MC] Text from [AR] added to Section 3.1.2.*

**[RC] P11 Line 16: Is it really clear or fair to say that the parameters related to**

heterotrophic respiration are particularly important when you have introduced a new function for the impact of soil moisture on respiration which has quite a number of parameters? Is it surprising that you have more equifinality for heterotrophic respiration given this new function? It would be good to describe this more in the results because at this point I was wondering if this was a clear conclusion of the analysis (i.e. that the heterotrophic respiration-related parameters are particularly important). A link to a figure at this point in the discussion may also help.

[AR] We agree that it is not clear that the parameters related to heterotrophic respiration are more important for Australian NEP than other parameters, and that the text "particularly those important for heterotrophic respiration" should be removed. However, we note that the number of parameters in this function is not an issue, as of the six parameters in this function, two parameters are not used at all (w2 and w3) so will not effect model outputs. And the function is reasonably well constrained by observations as discussed elsewhere. Prior to this work, it would not have been possible to assess the importance of parameter uncertainties on the soil respiration fuction because the function was fixed.

[MC] Remove "particularly those important for heterotrophic respiration". Add to Section 3.1.1 "Parameters $w_2$ and $w_3$ are unconstrained, but due to the values of the other parameters in this function they are not used they so are irrelevant to the model."

[RC] P11 Lines 21-22: Similar point to above. It is not immediately obvious to me, looking at Figure 9, S14 and S15 that the NEP anomaly is dominated by equifinality in the heterotrophic respiration. It depends which ecosystem contributes to the anomaly, which was only briefly discussed. Looking at Australia overall for 2011, it looks like the uncertainty due to parameter equifinality is the roughly the same for NPP and heterotrophic respiration. I think the same can be said for the ecosystems the authors say below contribute most to the NEP anomalies (savanna and sparse vegetation).

[AR] Yes, the reviewer is right. The uncertainty due to equifinality is larger for heterotrophic respiration than NPP in the tropics and temperate regions, but they are similar for Australia as a whole, or for the savanna and sparsely vegetation regions that contribute most to Australian NEP.

[MC] In the Abstract and Conclusions change text to "with similar contributions from equifinality in parameters associated with NPP and heterotropic respiration." Change text in Figure 3.2 to "The range in heterotrophic respiration is larger than the range in NPP in the tropics and temperate regions, but they are similar in other regions and for Australia as a whole (Figs. S14 and S15)."

[RC] In general all the routines used from PEST should be described in more detail somewhere (supplementary?).

[AR] There is documentation of the PEST routines in the PEST manual, so we do not believe it is worth describing them in detail, however we can add a short summary of what each routine does. The routines that we used were RANDPAR, PNULPAR, PARREP for null space Monte Carlo, and IDENTPAR for identifiability. We have already mentioned that GENLINPRED was used to calculate observation worth.

[MC] We add to Section 2.4 "We use the parallel implementation of PEST called BEOPEST." and "(specifically, we used routines SUPCALC to calculate the dimension of the solution space, RANDPAR to generate random parameter sets, PNULPAR to retain the null-space components of

*the random parameter sets and replace the solution space components with that of the calibrated model, and PARREP to replace parameters in the model's control file with the new parameter sets in preparation for recalibration)". In Section 3.1.1 we add that parameter identifiability was calculated with routine IDENTPAR. We had already mentioned that observation worth was calculated with routine GENLINPRED.*

**Other changes**

Added Holgate et al reference at the end of Section 2.1.1.

We added some additional references from the OzFlux Special Issue (Isaac et al 2016, Beringer et al 2016b) to Section 2.3.

Added "exceptional" in the abstract for emphasis.

Added Tilo Ziehn acknowledgement.

Added an additional step to the NSMC description in Section 2.4 "(vi) Eliminate any parameter sets that are not considered plausible."

[revised manuscript text omitted]

---

## Author Response (AR2)

**Author response to re-reviewer comments and manuscript changes in "Interannual variability in Australia's terrestrial carbon cycle constrained by multiple observation types", by Trudinger et al**

We thank both reviewers for their careful consideration of the manuscript, and provide the following responses:

**Author response to comments by Referee #1**

**[RC] (1) P7 Line 21: Did the authors mean to say "null and solution" spaces instead of "null and calibration"?**

[AR] Yes, thank you. We will change "calibration" to "solution".

**Author response to comments by Referee #2**

**[RC] (1) I am not completely convinced by the author on the question of "why the authors used monthly fluxes instead of daily fluxes in optimization". The authors argued "Although the model produces daily carbon and water fluxes, we used monthly rather than daily flux observations because the meteorological forcing is not as reliable at the daily timescale". But how can you trust the monthly modeled fluxes which are aggregated from the "inaccurate" daily fluxes driven by the "inaccurate" meteorological forcing? I noticed that this is a common question raised by both reviewers, so I believe it could be a potential question mark for other readers too. It may not worth a very long discussion, but some more explanation may be necessary.**

[AR] We should have been clearer here. This relates specifically to the precipitation data, that is not as accurate at the daily timescale, but the daily values have been rescaled to be consistent with the more accurate monthly observations. We will change the text to "Although the model produces daily carbon and water fluxes, we used monthly rather than daily flux observations because the precipitation data are more reliable at monthly timescales (Jones et al., 2009), as follows: Precipitation is spatially very variable at daily timescales, thus difficult to observe accurately with the relatively sparse gauge network. The spatial pattern of precipitation at monthly timescales is significantly smoother, therefore more accurately interpolated between the measurement locations. The daily precipitation data we use in the model has been rescaled (Jones et al., 2009), so that the sum of the daily values over a month is consistent with the interpolated monthly values."

**[RC] 2. How do you avoid local optimum when you use down-gradient method for optimization?**

[AR] We will add the following paragraph to the end of Section 2.4 to explain: "Down-gradient methods such as the Gauss-Marquardt-Levenberg method have the important advantage of being much more computationally efficient than global search methods. However, they can suffer from the disadvantage of finding only a local minimum, and not the global minimum (Raupach et al., 2005; Wang et al., 2009). There are a number of ways to significantly reduce the chances of getting stuck in a local minimum with PEST (Doherty, 1999). One way is by the choice of the parameter increments in the different stages of the optimsation (large increments at first, then smaller increments), and how the derivatives are calculated. Another way is by the choice of the initial parameter values (as good as possible, such as based on expert knowledge) or by repeating the optimisation with different initial parameter values. The use of the NSMC method allows

a much more thorough search of parameter space than a single PEST optimisation, but in a computationally efficent way. Skahill and Doherty (2006) describe additional ways to improve the chances of finding the global minimum with PEST. Here, we reduce the chances of PEST getting stuck in a local minimum by basing our choices for parameter increments and the derivative calculation method on recommendations in Doherty (1999), and by using the NSMC method."

**Other changes:** Fixed a couple of spelling errors, updated a couple of references that are now published and made minor changes to a couple of figure captions.

[revised manuscript text omitted]